



# Future trends in stratosphere-to-troposphere transport in CCMI models

Marta Abalos[1], Clara Orbe[2], Douglas E. Kinnison[3], David Plummer[7], Luke D. Oman[6], Patrick Jöckel[5], Olaf Morgenstern[4], Rolando R. Garcia[3], Guang Zeng[4], Kane A. Stone[8, 9, *], and Martin Dameris[5]

[1]Department of Earth Physics and Astrophysics, Universidad Complutense de Madrid, Madrid, Spain
[2]NASA Goddard Institute for Space Studies, New York, NY, USA
[3]National Center for Atmospheric Research, Boulder, CO, USA
[5]Deutsches Zentrum für Luft- und Raumfahrt (DLR), Institut für Physik der Atmosphäre, Oberpfaffenhofen, Germany
[4]National Institute of Water and Atmospheric Research (NIWA), Wellington, New Zealand
[6]NASA Goddard Space Flight Center, Greenbelt, MD, USA
[7]Climate Research Branch, Environment and Climate Change Canada, Montreal, Canada
[8]School of Earth Sciences, University of Melbourne, Melbourne, Victoria 3010, Australia
[9]ARC Centre of Excellence for Climate System Science, University of New South Wales, Sydney, New South Wales 2052, Australia
[*]Now at: Department of Earth, Atmospheric and Planetary Sciences, Massachusetts Institute of Technology, Cambridge, MA, USA

**Correspondence:** Marta Abalos (mabalosa@ucm.es)

**Abstract.** One of the key questions in the air quality and climate sciences is how will tropospheric ozone concentrations change in the future. This will depend on two factors: changes in stratosphere-to-troposphere transport (STT) and changes in tropospheric chemistry. Here we aim to identify robust changes in STT using simulations from the Chemistry Climate Model Initiative (CCMI) under a common climate change scenario (RCP6.0). We use two idealized stratospheric tracers to isolate changes in transport: stratospheric ozone ($O_3S$), which is exactly like ozone but has no chemical sources in the troposphere, and st80, a passive tracer with fixed volume mixing ratio in the stratosphere. We find a robust increase in the tropospheric columns of these two tracers across the models. In particular, stratospheric ozone in the troposphere is projected to increase 10-16% by the end of the 21st century in the RCP6.0 scenario. Future STT is enhanced in the subtropics due to the strengthening of the shallow branch of the Brewer-Dobson circulation (BDC) in the lower stratosphere and of the upper part of the Hadley cell in the upper troposphere. The acceleration of the deep branch of the BDC and changes in eddy transport contribute to increase STT at high latitudes. The idealized tracer st80 shows that these STT changes are dominated by greenhouse gas (GHG) increases, while phasing out of ozone depleting substances (ODS) does not lead to robust STT changes. Nevertheless, the increase of $O_3S$ concentrations in the troposphere is attributed to GHG only in the subtropics. At middle and high latitudes it is due to stratospheric ozone recovery linked to ODS decline. A higher emission scenario (RCP8.5) produces qualitatively similar but stronger STT trends, with changes in tropospheric column $O_3S$ more than three times larger than those in the RCP6.0 scenario by the end of the 21st century.



# 1  Introduction

Ozone is most abundant in the stratosphere, and its presence is crucial for protecting life on Earth from the harmful solar ultraviolet radiation. In the troposphere, ozone acts as a greenhouse gas and near the surface as a toxic pollutant (e.g. Ramaswamy et al. (2001), WHO (2003)). Because the stratosphere can be regarded as a reservoir of ozone, changes in stratosphere-to-troposphere transport (STT) play a very important role in determining the evolution of tropospheric ozone (Zeng and Pyle (2003), Collins (2003), Sudo et al. (2003), Zeng et al. (2010)). The future evolution of tropospheric ozone concentrations remains highly uncertain. A significant part of the uncertainty is due to the climate change scenario, in particular the projected changes in ozone precursor emissions (Stevenson et al. (2013)). Specifically, the global burden of tropospheric ozone is estimated to decrease in the RCP6.0 scenario of the Intergovernmental Panel on Climate Change (IPCC) (Sekiya and Sudo (2014), Revell et al. (2015)), and is expected to increase in the RCP8.5 scenario (Banerjee et al. (2016); Meul et al. (2018)) over the 21st century. There are also significant uncertainties for a specific future scenario due to differences between models (Dhomse et al. (2018), Morgenstern et al. (2018)).

Although there is a large uncertainty related to the evolution of chemical precursors of ozone (e.g. WMO (2018)), changes in STT are expected to make an important contribution to future tropospheric ozone changes (e.g. Hegglin and Shepherd (2009), Kawase et al. (2011)). The 2018 WMO Ozone Assessment reports that models project future increases in STT of ozone, but the magnitude of the change is strongly scenario-dependent, and there is no multi-model study to date (Karpechko et al. (2018)). The enhancement of STT is generally attributed to the acceleration of the Brewer-Dobson circulation (BDC) which is predicted consistently by climate model simulations in response to increasing greenhouse gases (Butchart and Scaife (2001)). This enhanced circulation leads to stronger downwelling, and thus to accumulation of ozone in the extratropical lowermost stratosphere, often referred to as the "middle world", thereby increasing the ozone reservoir available for transport into the troposphere. Two branches of the BDC are usually considered, the deep branch with downwelling confined to polar latitudes (poleward of ∼60°N/S) and the shallow branch with downwelling over subtropics and midlatitudes (Birner and Bönisch (2011)).

The amount of stratospheric tracer (e.g. ozone) transported into the troposphere will depend on the frequency of cross-tropopause irreversible transport events as well as on the concentration in the lower stratosphere reservoir (Albers et al. (2017)). The latter is controlled by changes in the BDC, in addition to chemical production and loss in the "middle world". The cross-tropopause irreversible transport occurs typically through isentropic mixing around mid-latitude disturbances in the Atlantic and Pacific storm tracks, and some times associated with tropopause folds (e.g. Stohl et al. (2003)). Therefore, STT is frequently caused by Rossby wave breaking near the tropopause (Škerlak et al. (2014), Boothe and Homeyer (2017), Yang et al. (2016)). In this paper we will address the transport mechanisms leading to the STT increases from the Transformed Eulerian Mean (TEM) perspective. This methodology provides novel insights into the STT mechanisms and their future changes, as it allows evaluating the contributions to the net transport from advective transport by the mean meridional circulation and two-way mixing. In addition, by using idealized tracers with stratospheric sources implemented in the models, we are able to evaluate long-term changes in net STT without having to estimate the flux indirectly from the budget in the "middle word".





Previous studies have obtained estimates of the future changes in STT. Butchart and Scaife (2001) estimate an increase in STT of about 3%/decade, and highlight the important consequences on the rate of chlorofluorocarbon (CFC) removal from the stratosphere. Based on correlations over the observational period between the residual circulation and mid-tropospheric ozone, Neu et al. (2014) estimate an increase in zonal-mean tropospheric ozone concentrations of 2% by the end of the 21st century due

to an enhanced BDC. Hegglin and Shepherd (2009) estimate the change in ozone STT flux and obtain a 23% increase from 1965 to 2095 due to climate change in a chemistry-climate model forced with A1B emissions scenario (Nakicenovic et al. (2000)). More recent modeling studies have used artificial tracers to extract the changes due to STT from those due to tropospheric chemistry. Using a stratospheric ozone tracer with no chemical ozone production in the troposphere, $O_3S$, Banerjee et al. (2016) and Meul et al. (2018) provide the latest estimates of the future increases in the STT of ozone. In particular, Meul et al.

(2018) argue that ozone STT flux will increase more than 50% by 2100 under an RCP8.5 scenario.

In the present study we examine future trends in STT from a multi-model perspective using a subset of the CCMI models that provide the necessary output. Section 2 presents the models and tracer output used, Section 3 shows the 21st century trends in the tracers and Section 4 examines the associated changes in transport mechanisms. Section 5 compares the results for two IPCC scenarios, RCP6.0 and RCP8.5, and examines the separate contributions to the STT trends from greenhouse gases (GHG)

and ozone-depleting substances (ODS). Section 6 summarizes the main conclusions of the study.

## 2   Data and Method

We use model output from the CCMI project for seven models over the period 2000-2100. Specifically, we use the REF-C2 simulations as the control, which have time-varying emissions that follow the RCP6.0 IPCC scenario. In Section 5 we use additional sensitivity simulations, including the SEN-C2-RCP85, using the RCP8.5 IPCC scenario, and also SEN-C2-fODS

and SEN-C2-fGHG simulations. The last two sensitivity simulations are exactly the same as the REF-C2 simulations, but ozone-depleting substances (ODS) or greenhouse gases (GHG) are fixed to 1960 levels, respectively. This allows attribution of the trends observed in REF-C2 to either external forcing. Morgenstern et al. (2017) provides a description of the CCMI models and simulations, as well as the references for each model. Our analyses are focused on the use of the idealized tracers $O_3S$, st80 and e90, which are described in Eyring et al. (2013). $O_3S$ is the same as $O_3$ in the stratosphere and it decays chemically in

the troposphere, but it is not produced in this layer. The tracer st80 is continuously set to a specified constant mixing ratio everywhere at 80 hPa and above. Outside this region it is a passive tracer, and in the troposphere it decays with a 25 day e-folding timescale. In addition to these stratospheric tracers, the tropospheric tracer e90 is used. This tracer is emitted throughout the surface (constant mixing ratio boundary condition) and decays everywhere in the atmosphere with a 90-day lifetime. Table 1 lists the model output used in this study, including the available tracer fields. Orbe et al. (2018) examined tropospheric trans-

port using some of these tracers, and reported some implementation issues, which are included in Table 1. For instance, in the EMAC model the st80 tracer decays everywhere below 80 hPa, instead of only in the troposphere. For ACCESS and NIWA no known issues have been detected in the implementation of st80 but, as will be shown below, the behavior of the idealized tracer is inconsistent with changes in transport and in $O_3S$. As will be seen below, the magnitude of $O_3S$ and the fraction of ozone in





[H]

**Table 1.** Table 1. Available tracer model output for the REF-C2 simulations. ✓ available, ✗ not available, ✳ implementation issues.

|            | ACCESS | CMAM | EMAC-L47MA | EMAC-L90MA | GEOSCCM | NIWA | WACCM |
|------------|--------|------|------------|------------|---------|------|-------|
| $O_3$      | ✓      | ✓    | ✓          | ✓          | ✓       | ✓    | ✓     |
| $O_3S$     | ✓      | ✓    | ✓          | ✓          | ✓       | ✓    | ✓     |
| st80       | ✳      | ✓    | ✳          | ✳          | ✓       | ✳    | ✓     |
| e90        | ✗      | ✓    | ✗          | ✗          | ✓       | ✗    | ✓     |

the troposphere that was accounted for by $O_3S$ varies considerably between models. All models implemented $O_3S$ similarly, with $O_3S$ loss in the troposphere defined as the photochemical loss of ozone including the effects of dry deposition. However the details of which chemical reactions were defined as contributing to photochemical ozone loss, as opposed to recycling, varied between models and explains part of the spread in the magnitude of $O_3S$. As our primary interest is understanding the

long-term trends in $O_3S$ and trends in the contribution of $O_3S$ to tropospheric ozone, we do not further consider the differences in the magnitude of $O_3S$ across the models. Note that the NIWA and ACCESS models have the same atmospheric model, but ACCESS has prescribed ocean from CMIP5 HadGEM2-ES while NIWA has an interactive ocean (Morgenstern et al. (2017)). In addition, we note that the coupled ocean version of EMAC-L47MA is used here, while the EMAC-L90MA simulation has prescribed sea surface temperatures. The SEN-C2-RCP8.5 simulations are only available for CMAM, EMAC-L47MA and

WACCM, while the SEN-C2-fODS and SEN-C2-fGHG are available for ACCESS, CMAM, NIWA and WACCM.

The transport changes underlying the changes in stratospheric tracer concentration in the troposphere will be examined through the analysis of the TEM budget. The TEM tracer continuity equation can be written for the zonal mean tracer mixing ratio $\bar{\chi}$ on pressure levels as (Andrews et al. 1987)

$$\bar{\chi}_t = -\bar{v}^*\bar{\chi}_y - \bar{w}^*\bar{\chi}_z + \nabla \cdot M + \bar{P} - \bar{L} + \bar{X} \qquad (1)$$

where $M = -e^{-z/H}\left(\overline{v'\chi'} - \frac{\overline{v'T'}}{S}\bar{\chi}_z, \overline{w'\chi'} + \frac{\overline{v'T'}}{S}\bar{\chi}_y\right)$ is the eddy tracer flux vector. The subscripts indicate partial derivatives, $(\bar{v}^*, \bar{w}^*)$ are the residual circulation components, overbars indicate zonal mean and primes deviations from it, $S = HN^2/R$, with the scale height $H = 7$ km and $R$ is the ideal gas constant, and the log-pressure altitude is $z = H ln(p_0/p)$, with $p_0 = 1000$ hPa. The chemical net tendency is given by the production minus loss term $\bar{P} - \bar{L}$. The resolved transport terms

describe advection by the residual circulation $-\bar{v}^*\bar{\chi}_y - \bar{w}^*\bar{\chi}_z$ and eddy transport $\nabla \cdot M$, related to two-way mixing. In addition to these resolved terms there is unresolved or subgrid scale transport, which includes numerical diffusion and parameterized processes such as gravity waves and convection ($\bar{X}$). However, here we will focus on the main resolved transport terms, and we refer the reader to Abalos et al. (2017) for a detailed discussion of the other terms in the budget in WACCM. We note that daily mean data is needed to compute the TEM terms, for both the dynamical and chemical fields. This is not available in all

models for the artificial tracers, so we only present the TEM budgets for which daily output was available.





## 3 Robust increase in future STT

### 3.1 Timeseries of tracer concentrations

Figure 1 shows timeseries over the 21st century of tropospheric columns of ozone (Fig. 1a), stratospheric ozone $O_3S$ (Fig. 1b) and the artificial tracer st80 (Fig. 1c), as well as the ratio of stratospheric to total ozone $O_3S/O_3$ (Fig. 1d) for the REF-C2
simulations. The tropospheric tracer columns are based on the thermal tropopause, which is provided as an output for every model. Overall, this figure demonstrates that the concentration of stratospheric tracers in the troposphere will increase in the future, and this result is robust across the CCMI models. The total ozone concentration in the troposphere (Fig. 1a) increases in most models until the middle of the century, and then decreases (except ACCESS and NIWA which show a near constant decrease throughout the century). Comparing the total ozone to the stratospheric ozone evolution (Figs. 1a and 1b) it is clear
that the future evolution of chemical production in the troposphere is crucial for the future ozone concentrations. Although not shown here, we have confirmed that a sensitivity WACCM simulation with fixed tropospheric ozone precursor emissions (SEN-C2-fEmis) does not show a decrease in the second half of the century as that seen in Fig. 1a. In the IPCC scenario RCP6.0 under consideration, methane emissions increase until 2080 and then decrease, and nitrogen monoxide emissions decrease since 2000 and more rapidly in the second half of the century (not shown, see Meinshausen et al. (2011) and IPCC report
2013). The evolution of these gases is likely contributing to the tropospheric ozone column behavior in Fig. 1a. In contrast, Fig. 1b shows that stratospheric ozone ($O_3S$) in the troposphere increases monotonically thoroughout the century, with a slight reduction of the concentrations over the two last decades in most models. This small reduction is linked to photochemical production of ozone from methane in the stratosphere, as we have confirmed examining additional sensitivity runs of the model CMAM with methane emissions fixed to 1960 levels (not shown). The fraction of stratospheric to total ozone in the troposphere
increases linearly in all models (Fig. 1d). In this sense, the models show a good qualitative agreement, although the ratio shows a large spread, from 22 to 45% for the year 2000. Trends in the ratio range from 0.32 to 0.99 %/decade among the models, with CMAM showing the largest trends and GEOSCCM the smallest. Note that the $O_3S$ increase is also seen in ACCESS and NIWA, which combined with the near flat trends in $O_3$, implies an increase in the stratospheric ozone fraction.

For st80 (Fig. 1c) the timeseries shown are standardized anomalies, i.e. the climatology is subtracted and the result is divided by the year-to-year standard deviation of the full timeseries, in order to focus the attention on the temporal evolution and not on the different magnitude of the concentration across the models. For instance, the magnitude of this artificial tracer in the EMAC models is notably smaller than in the rest (about 50 times), due to an implementation issue (see Section 2). However, the standardized anomalies show comparable values in Fig. 1c. The artificial tracer st80 shows an increase over the 21st
century in all models except for NIWA and ACCESS. The evolution of st80 is very different in each of these two outliers, and cannot be reconciled with the increase seen in $O_3S$ in both models. For this reason in the rest of the paper the st80 tracer will not be considered in these two models. As mentioned above, these two are essentially the same model. We emphasize here that st80 allows directly attributing this increase to enhanced STT, excluding the contribution of changes in stratospheric ozone chemistry present in $O_3S$. Specifically, an increase in stratospheric ozone is expected from stratospheric cooling and the




continued decrease in halogens over the 21st century. For instance, the flattening of $O_3S$ timeseries towards the end of the 21st century linked to the evolution of methane in this scenario is not seen in st80.

Figures 2a and 2b show the trends in $O_3S$ and st80 tropospheric columns as a function of the climate response of each model, estimated as the rate of warming of the tropical upper troposphere (30S-30N, 400-150 hPa) under the RCP6.0 scenario.

Note that we avoid using the term climate sensitivity because these simulations include several forcings in addition to $CO_2$, including, for example, time-varying ozone depleting substances (ODS). EMAC-L47MA (coupled ocean), GEOSCCM and WACCM have the smallest climate responses; EMAC-L90MA (prescribed ocean) and CMAM the largest. Although there is some spread, a relation between the two variables can be observed: models with larger climate response tend to produce larger trends in STT, and this relation is more clearly seen for st80 than for $O_3S$. Note that st80 trends are computed for the

standardized timeseries, as shown in Fig. 1. The statistical significance of all trends in the paper is computed with a two-tails Student t test for the slope at the 95% confidence level (Storch and Zwiers (1999)).

Abalos et al. (2017) showed a strong coupling between changes in upper troposphere and lower stratosphere (UTLS) transport and changes in tropopause height. The tropopause is expected to rise in a warming climate due to thermodynamical processes (Vallis et al. (2015)). An important role of the tropopause rise for the BDC trends has been also pointed out recently

by Oberländer-Hayn et al. (2016). Figure 2e shows a positive correlation between the change in global mean tropopause altitude and the climate response across the models. This means that models with higher climate response have larger tropopause height rises. Based on these findings, we may hypothesize that the tropopause changes act as a mediator between climate response and STT trends. However, Figures 2c and 2d show that the correlations between STT and tropopause rise are weaker than those between STT and climate response, implying that the relation is not simple.

In addition to the stratospheric tracers considered above, the evolution of a tropospheric artificial tracer provides a complementary view of the trends in stratosphere-troposphere exchange (STE). Figure 3 shows trends in the tropospheric tracer e90 in the troposphere for the three models that provide this output (CMAM, GEOSCCM and WACCM). The magnitude of the tracer trends increases with the climate response of the model. Note that there is a non-significant trend in WACCM, different from Figure 13 in Abalos et al. (2017), which showed a significant decrease of the e90 concentrations averaged over the troposphere

(in ppbv). However we note that the difference in the trends in the tropospheric average in Abalos et al. (2017) versus those in the tropospheric column in Fig. 3a is not significant, since the 95% confidence intervals overlap between the two. The other two models show a significant net decrease, larger for CMAM. This reduction of tropospheric tracer concentrations in the troposphere is consistent with a more efficient transport into the stratosphere, and is related primarily with stronger tropical upwelling in the future (e.g. Rind et al. (2001), Abalos et al. (2017)).

## 3.2 Spatial structure of the tracer trends

As a first step to understand the changes in transport processes leading to long-term trends in the STT rates seen in Figs. 1-3, in this Section we examine the zonal mean spatial structure of the trends. We note that the trends in EMAC-L90MA are qualitatively very similar to those in EMAC-L47MA and thus only the latter is shown in the rest of the paper, denoted EMAC.





Starting with the tropospheric tracer, Figure 4 shows the spatial structure of the trends in e90. The three models yield highly consistent results, and the magnitude scales with the climate response shown in Fig. 3. Using the TEM formalism, Abalos et al. (2017) examined the causes leading to these trend patterns. In the stratosphere, the increase in the tropics is attributed to enhanced BDC tropical upwelling, while the increase near the extratropical tropopause is due to enhanced isentropic mixing

in this region. The rise of the tropopause altitude plays a key role in these stratospheric changes, shifting upward the region of wave dissipation, and thus isentropic mixing and the residual circulation wave forcing (Abalos et al. (2017)). In the extratropical troposphere, the negative trends are associated with weaker meridional mass circulation and isentropic mixing. In the tropical troposphere, they are due to enhanced and deeper circulation in the upper part of the Hadley cell, linked to enhanced deep convection and connected to the stronger BDC tropical upwelling. In simple terms, the reduction in the tropospheric e90

concentrations reflects a less efficient transport from the surface into the extratropical free troposphere and a more efficient troposphere-to-stratosphere transport in the future.

The trends in $O_3S$ are shown in Figure 5. In the stratosphere, ozone decreases in the tropics and increases in the extratropics, and this pattern is expected from the acceleration of the residual circulation seen in all models (as will be shown below). There is an interhemispheric asymmetry in the lowermost stratosphere, with positive trends in the SH extending to lower levels than

in the NH. The large SH $O_3S$ positive trends are due to the recovery of the Antarctic ozone hole. In contrast, the negative trends around the NH extratropical tropopause are consistent with trends in the artificial tracer e90 identified in Fig. 4. As stated above, these trends around the extratropical tropopause are attributed to changes in isentropic mixing in this region, and constitute a fingerprint of the tropopause rise (see Abalos et al. (2017)). This is confirmed by looking at the $O_3S$ trends in tropopause-relative coordinates, in which the band of negative trends at the NH tropopause is not present (not shown). The

change in tropopause altitude as a function of latitude over the 21st century is shown in Figs. 4, 5 and 6. In the troposphere, $O_3S$ trends are positive, and their spatial structure reveals a common pattern across models (Fig. 5). The largest trends are found in the subtropics, near 30N/S in the middle to upper troposphere (∼500-200 hPa). The positive trends extend to higher latitudes below the tropopause, and these extratropical increases are stronger in the SH than in the NH in all models. The deep tropics display a local minimum and near zero trends or even negative are found next to the ground in the tropics, consistent

with little influence of stratospheric air in these regions. This pattern is consistent with those shown by Banerjee et al. (2016) and Meul et al. (2018) in individual models.

In Figure 6 we use the artificial tracer st80 in order to examine the trend patterns for an artificial tracer with constant and homogeneous stratospheric sources. Note that this tracer is only useful in the troposphere, since it has a constant fixed value of 200 ppbv at and above 80 hPa, and thus the stratospheric values are not shown. The st80 trends in EMAC (Fig. 6b) have

been multiplied by an arbitrary factor of 50 in order to fit the scale due to a known implementation issue (see Section 2). The st80 trends in Fig. 6 show more hemispherically symmetric patterns as compared to Fig. 5, with negative trends near the extratropical tropopause in both hemispheres (except in EMAC), as seen in the NH trends for $O_3S$ and consistent with e90 trends in Fig. 4. In addition, there are positive trends maximizing in the subtropical troposphere in both hemispheres, that extend to higher latitudes below the region of negative trends linked to the extratropical tropopause. This structure is similar

to that seen for $O_3S$, although not exactly comparable due to differences in the gradients between these two tracers and their


different stratospheric source distributions. In the deep tropical upper troposphere all models show a local minimum, although CMAM and EMAC show higher concentrations at the equator than GEOSCCM and WACCM. The st80 concentrations in the deep tropics are likely linked to the altitude of the tropopause in the models. In particular CMAM shows the highest tropopause, with the tropical tropopause altitude close to the 80 hPa level in which st80 has its constant source (Fig. 6a). As the tropopause

rises in the simulated future climate, this can result in enhanced vertical diffusion into the tropical troposphere. Overall, the $O_3S$ and st80 trend patterns highlight common features in all models, and in the next section we investigate the transport changes that lead to these trends in the stratospheric tracers.

## 4    Trends in TEM transport terms

To facilitate interpretation of the TEM transport terms that will be shown next, Figure 7 shows the trends in the residual

circulation streamfunction for the various models included in this study. In the stratosphere, all models show an acceleration of the BDC in both hemispheres. However, while in the NH the acceleration extends from the tropics to polar latitudes, in the SH there is a cell of opposite sign at high latitudes. This is due to the impact of the ozone hole recovery on the residual circulation. Specifically, the ozone recovery weakens the downwelling in the SH high latitudes over the 21st century, reversing the acceleration induced during the ozone hole formation over the las decades of the 20th century (Polvani et al. (2018),

Abalos et al. (2019)). This behavior is observed in most CCMI models (Morgenstern et al. (2018), Polvani et al. 2019). Figure 7 confirms this common behavior, with EMAC being an exception with near zero residual circulation trends in the SH polar latitudes. The fact that this feature does not appear as clearly in this model could be linked to a weak SH polar vortex.

In the troposphere, most models show a deceleration of the residual circulation in the extratropics of both hemispheres. In the tropics, all models show a strengthening of the upper part of the Hadley cell, just below the acceleration of the shallow

branch of the stratospheric circulation. As shown in Abalos et al. (2017), this strengthening is closely linked to the upward shift of the tropopause. As will be shown below, the enhanced downwelling of the Hadley cell in the upper subtropical troposphere cooperates with the enhanced shallow branch of the BDC to enhance advective downward transport of the stratospheric tracers into the troposphere. This leads to the subtropical upper tropospheric maxima in the $O_3S$ and st80 trend patterns seen in Figs. 5 and 6. Moreover, Figure 7 demonstrates that this mechanism is present in all models.

Figures 8 and 9 show the 21st century change in the advective transport term for the tracers $O_3S$ and st80, respectively, in four models (CMAM, EMAC, GEOSCCM and WACCM). We note that the daily output for CMAM was obtained three times per month, while the other models had output for every day of the year. In order to increase the amount of data in CMAM, we considered differences between the first and last half of the century for this model. For the others, differences between the first

and last 10 years of the century are considered.

The $O_3S$ advective term (Fig. 8) shows a decrease in concentrations in the tropical lower stratosphere due to enhanced upwelling, and increases at extratropical latitudes due to enhanced downwelling in the deep branch of the BDC. While in the NH the extratropical increases extend into polar latitudes, in the polar SH there are negative ozone trends by advection above 20



km in all models except EMAC. This is due to the impact of ozone recovery on the residual circulation highlighted above, with downwelling weakening over the 21st century in this region (shown in Fig. 7). Interestingly, in the SH polar lower stratosphere below 20 km, the ozone recovery due to decreasing ODS concentrations compensates this reduction in downwelling, and the net effect is to increase the ozone downward transport in that region. As a result, there is enhanced advective transport of

stratospheric ozone into the troposphere at polar latitudes in both hemispheres. In addition, all models show positive trends in ozone due to advection near the subtropical tropopause (approximately 30N and 30S at 150 hPa). The arrows indicate an enhancement of the amount of stratospheric ozone being advected across the tropopause into the troposphere in these regions by the enhanced shallow branch of the BDC and top of the Hadley cells. These patterns lead to the subtropical tongues observed in the $O_3S$ trends in Fig. 5. Hence, both shallow and deep branches of the BDC lead to enhanced stratospheric ozone transport

into the troposphere.

Figure 9 shows the advective transport term for the artificial tracer st80 in the same four models. Again, all models show consistent features. In particular, similar to the behavior seen in $O_3S$, there is enhanced downward transport into the tropo-sphere in the subtropics linked to the shallow BDC and top of the Hadley cell. These patterns lead to the subtropical tongues in the st80 trends seen in Fig. 6. At polar latitudes only GEOSCCM and WACCM show enhanced downward transport into

the troposphere in the NH hemisphere. This is due to the enhanced downwelling of the deep branch as seen for $O_3S$. The fact that CMAM and EMAC-L47MA fail to capture these trends for st80 while showing them for $O_3S$ suggests that there might be issues with the st80 idealized tracer in the stratosphere (this is indeed the case for EMAC, see Table 1). On the other hand, the fact that none of the models show enhanced downward transport of st80 at SH polar latitudes confirms that the enhanced $O_3S$ transport in this region seen in Fig. 8 is exclusively linked to ozone recovery.

While we find good correspondence among models in the advective transport term (Figs. 8 and 9), the eddy mixing term is subject to larger uncertainties and there is a larger spread among models. This term is computed from meridional and vertical tracer eddy fluxes (see Eq. 1 and related discussion) which are highly sensitive to numerical errors arising from different sources including limited temporal resolution of the output, implicit diffusion in the model transport scheme and vertical interpolations

(see Abalos et al. (2017) for a discussion of these errors in WACCM). Since understanding in detail the sources of uncertainty in each model is beyond the scope of the study and our goal is to extract robust trend features in the transport terms, we only include results for models showing consistent behavior. At the same time, we warn that the conclusions extracted for the eddy term should be considered more uncertain than those found for the advective term, given the limited inter-model agreement.

Figure 10 shows the trends in the eddy term for $O_3S$ (a and b) in CMAM and WACCM and for st80 in GEOSCCM and

WACCM (c and d). In the stratosphere there is enhanced isentropic mixing transporting $O_3S$ from high latitudes into the tropics in both models, and enhanced eddy transport out of the SH polar stratosphere. Of greater interest for this study are the trends in cross-tropopause transport. For both tracers there is enhanced isentropic mixing across the extratropical tropopause, which leads to increases in ozone concentrations in the upper troposphere middle latitudes in both hemispheres (30-60N/S). In addition to increasing the tropospheric concentrations, this enhanced eddy transport decreases the tracer concentrations

in the lowermost extratropical stratosphere. This is consistent with the enhanced isentropic eddy transport in e90 found by



Abalos et al. (2017) leading to the increased concentrations in this tracer around the extratropical tropopause (Fig. 4). Hence, the same mechanism leads to the negative trends in st80 and $O3S$ (only in the NH) around the extratropical tropopause (Figs. 5 and 6). Finally, there is enhanced cross-tropopause near-vertical transport of st80 in the tropics, which contributes to enhance the concentrations in the tropical upper troposphere on both sides of the equator. This feature is mainly due to the fixed st80
concentrations above 80 hPa: as the tropical tropopause rises and gets close to this level, vertical gradients strengthen and lead to cross-tropopause transport.

Overall, Figures 8 to 10 suggest that the trend patterns seen in $O_3S$ and st80, specifically the subtropical tongues, are largely explained by the enhanced advective transport into the troposphere, and not by changes in two-way mixing. The relevance of the mean advective circulation for the STT trends is a novel result that has not been considered before.

**5   Dependence on emission scenario**

These results shown up to now apply for the RCP6.0 scenario of the IPCC but, as mentioned in the Introduction, other scenarios may lead to different results. Figure 11 shows timeseries of tropospheric columns of various stratospheric tracers over the 21st century as in Fig. 1 but for the RCP8.5 scenario. This data is only available in three models: CMAM, EMAC-L47MA and WACCM. Comparing Fig. 11a to Fig. 1a reveals a clear difference in the evolution of ozone in the troposphere, which
increases throughout the century in the RCP8.5 scenario in the three models. Part of the difference in ozone could be explained by differences in the precursor emissions between the two scenarios. Methane increases monotonically throughout the century in the RCP8.5 scenario, while in the RCP6.0 scenario it increases at a much slower rate and then it decreases during the last two decades of the century (Meinshausen et al. (2011)). These differences in precursor (methane) emissions could explain the different ozone trends between the two scenarios. In contrast, nitrogen monoxide emissions decrease over the 21st century in
the two scenarios, with similar slopes (not shown), so they cannot explain the differences. In addition to these chemical effects, the strong acceleration of the BDC in the more extreme scenario leads to enhanced ozone STT as compared to the other scenario. Hence, we suggest that both chemical and dynamical effects are contributing to the faster increase in tropospheric ozone.

Consistent with a stronger increase in ozone STT, $O_3S$ increases faster in the more extreme scenario (Fig. 11b). A clear
difference between the two scenarios for this tracer is seen in the last 20 years of the simulations, where $O_3S$ concentrations remain flat in RCP6.0 but continue to increase in the RCP8.5 scenario. As mentioned in Section 3.1, this flattening of $O_3S$ in RCP6.0 is related to ozone production from methane in the lower stratosphere. We note that this effect also likely contributes to the faster increase in $O_3S$ in RCP8.5 especially in the first half of the century, when the differences in BDC strength between the two scenarios are not as large (not shown). Finally, there could be a contribution from the different rate of stratospheric
ozone recovery due to larger stratospheric cooling in RCP8.5 (WMO (2018)). The ratio of stratospheric to total ozone (Fig. 11d) also increases faster in this extreme scenario than in the RCP6.0. For st80 (Fig. 11c) there is a steady increase in the RCP8.5 for all models. In CMAM and EMAC-L47MA this increase is enhanced over the last 20 years, perhaps due to the tropical tropopause being close to 80 hPa by the end of the century. The 21st century trends in $O_3S$ and st80 in the two scenarios are





compared in Figs. 11e and 11f. In all models the climate response is stronger in the higher emissions scenario. The tracer trends in general are significantly stronger in the RCP8.5 scenario, consistent with more severe climate change and associated transport changes. An exception is found for st80 in CMAM, which shows statistically undistinguishable trends in both scenarios. This could be linked to the tropical tropopause being at nearly 80 hPa in this model for both scenarios by the end of the century.

Polvani et al. (2018) showed that the future acceleration of the BDC due to the increase in GHG is partly compensated by the decrease in ozone-depleting substances (ODS). More specifically they showed that the global mean age of air trends in the 21st century are reduced by about half with respect to those observed over the last few decades of the 20th century. The main reason for this decrease is the weakening of the polar SH downwelling associated with the recovery of the ozone hole. Here,

we examine the separate contributions to the STT trends from changes in the GHG and ODS emissions. Banerjee et al. (2016) and Meul et al. (2018) examined the impact of these two forcings on ozone transport into the troposphere in individual models, and found that increases in GHG lead to increases in stratospheric ozone in the subtropics, while ozone recovery leads to ozone enhancement at higher latitudes.

Figure 12 shows the trends in $O_3S$ for the three models that output this tracer (ACCESS, CMAM and NIWA) for sensitivity

simulations with concentrations of ODS or GHG fixed to 1960 levels. Since we are looking at trends for the 21st century, the fixed-ODS simulations can be conceptualized as characterizing the impact of the increase of GHG through the century, and the fixed-GHG simulations as characterizing the impact of the elimination of halogens. Comparing Figs. 12a 12c and 12e with Fig. 5 it is clear that increases in stratospheric ozone in the troposphere are more modest when ODS concentrations are fixed. This simply reflects the fact that stratospheric ozone recovery leads to more ozone being transported into the troposphere. In

agreement with previous studies, in the fixed-ODS simulations the $O_3S$ increases are limited to the subtropical region. The negative trends near the extratropical tropopause are consistent with the changes in mixing associated with the tropopause rise in response to climate change mentioned above (Figs. 4-6, Abalos et al. (2017)). The differences between the control and sensitivity simulations are especially evident in the SH, where the ozone hole is located and thus the ozone recovery effect is strongest. This effect of ozone recovery is isolated in the fixed-GHG sensitivity simulations (Figs. 13b, 13d and 13f). These

clearly show that increases in stratospheric ozone concentrations result in increased STT mostly in the extratropics. The results in Fig. 13 are highly consistent with those from previous studies highlighted above (Banerjee et al. (2016) and Meul et al. (2018)).

In order to separate the effects of changes in transport from those of stratospheric ozone recovery we now look at st80 trends. Figure 13 shows trends in the sensitivity simulations for st80 in CMAM and WACCM (note that ACCESS and NIWA are not

used for the reasons explained in Section 3.1). In contrast with $O_3S$, the st80 trends in the fixed-ODS runs (Figs. 13a and 13c) are very similar to those in the control simulations for both models not only in the subtropics but also at high latitudes (Fig. 6). This confirms that the changes in transport mechanisms examined in the previous section are mainly caused by the increase in GHG. On the other hand, the trends in the fixed-GHG simulations (Figs. 13b and 13d) are much more uncertain and are statistically insignificant over large regions of the troposphere. For instance, the two models show opposite sign trends in

the NH (not significant in WACCM, and also not consistent among 3 different members, not shown). This difference between



the $O_3S$ and st80 trends in the fixed-GHG scenario reveals that the trends in $O_3S$ are mostly due to the increase in ozone concentrations, rather than by dynamical changes induced by ozone recovery (or ODS decrease). Nevertheless, one consistent feature in the two models is the increase in st80 concentrations in the SH high latitude upper troposphere (Figs. 13b and 13d). This is linked to a small downward shift of the tropopause in these simulations associated with the weakening of polar

downwelling, a dynamical effect of the recovery of the Antarctic ozone hole (Polvani et al. (2018), Polvani et al. 2019). Note that this SH tropopause shift is opposite to that associated with GHG increases (compare the tropopauses in Figs. 13a and 13c to Figs. 13b and 13d). Consistently, the positive st80 trends due to ozone recovery near the SH tropopause partly offset the decrease due to increasing GHG (the net result is still a decrease, as shown in Fig. 6). Besides this specific feature, no robust conclusion can be extracted on future changes in STT if GHG concentrations are fixed.

## 6   Conclusions

In this study we investigated the future trends in stratosphere-troposphere exchange (STE), with a specific focus on stratosphere-to-troposphere transport (STT), using output from various models participating in CCMI in order to extract robust signals. Idealized tracer data provides a unique opportunity to evaluate changes in transport, separated from changes in chemical processes due, for instance, to changes in precursor concentrations. We show that all models agree in predicting an enhancement

of the concentrations of stratospheric tracers in the troposphere in the future. The models project a near-linear increase in the fraction of stratospheric to total ozone in the troposphere between 10 and 20% over the 21st century with respect to the 2000-2010 mean values. This increase reflects the combined effects of increased transport - the tropospheric column of stratospheric ozone increases 10-16% - and reduced chemical production in the troposphere during the second half of the 21st century in the RCP6.0 emission scenario considered here. These results are consistent with previous studies using individual models. Our

multi-model approach allows extracting a correlation between the climate response and the magnitude of the STT trends in the different models. As shown in previous studies, tropopause altitude is a key player in the STE trends. While we obtain a positive correlation between tropopause rise in a future climate and climate response across models, a clear connection between tropopause rise and STT trends cannot be established.

    In addition to ozone and stratospheric ozone ($O_3S$), we analyze the idealized tracer st80, which is independent of chemistry

in the stratosphere as well as in the troposphere. The patterns of trends in $O_3S$ and st80 exhibit common features, revealing the fingerprints in the tracer trends of changes in transport alone (excluding changes in stratospheric ozone chemical sources and sinks). One of the key features identified in the tracer trends are maximum increases in the subtropical troposphere of both hemispheres. In addition, the positive trends extend to high latitudes in the lower troposphere (below approximately 400 hPa). Negative trends are generally observed around the extratropical tropopause, except for $O_3S$ in the SH, where stratospheric

ozone recovery leads to an increase in the concentrations across the tropopause.

    Examination of the TEM tracer continuity equation allows evaluation of the changes in transport processes leading to the enhanced STT, separating advective transport by the residual circulation from eddy transport, linked to two-way mixing. We find that enhanced advective transport plays a key role in the subtropical trend maxima for both $O_3S$ and st80. This is linked





to the acceleration of the shallow branch of the residual circulation and the strengthening of the top of the Hadley cell. Note that the Hadley cell strengthening does not extend into the lower troposphere (e.g. Fig. 7). This is consistent with recent studies arguing that metrics of the Hadley cell should be considered separately for the upper and lower parts (Davis and Davis (2018), Waugh et al. (2018)).

In the extratropics, we find enhanced downward advective transport by acceleration of the deep branch of the BDC in the NH only. In the SH the dynamical effects of ozone hole recovery prevail and there is a weakening of polar downwelling (Polvani et al. (2018)). Nevertheless, $O_3S$ STT increases due to the higher ozone concentrations in the lower stratosphere. Although larger uncertainties are obtained for the eddy transport term, a consistent feature identified is an enhancement of isentropic eddy transport near the extratropical tropopause. This leads to decreases in stratospheric tracer concentrations in

this region and contributes to the increases in the extratropical troposphere. Such enhanced mixing around the extratropical tropopause is also observed for the idealized tropospheric tracer e90, and it is linked to an upward shift of the extratropical tropopause and associated changes in the wave propagation conditions (Abalos et al. (2017)). We argue that the same mechanism leads to the negative trends in stratospheric tracers around the extratropical tropopause, which disappear when tropopause-relative coordinates are used.

While the same qualitative results apply when the more extreme scenario RCP8.5 is considered, the tropospheric column of stratospheric ozone increases 40-50% by the end of the 21st century (with respect to the 2000-2010 mean values). This implies an increase more than 3 times larger than in the RCP6.0 scenario. However, this result is limited to two models due to limited availability of output. The results for st80 also suggest a larger increase for the high emission scenario: two out of three models show 1.5 times larger trends in the RCP8.5 scenario. We also explore the dependence on the future emission

scenario and the roles of GHG and ODS separately. Using $O_3S$, Banerjee et al. (2016) and Meul et al. (2018) showed that increasing GHG lead to enhanced stratospheric ozone in the subtropical troposphere, while ozone recovery leads to more modest enhancements throughout the extratropics. Here we confirm that this is the case also for all the models providing the specific sensitivity simulations. Moreover, using the idealized tracer st80 we show that the robust STT trends are primarily attributed to increasing GHG emissions. Ozone recovery (or ODS reduction) does not drive consistent trends across models.

The assessment of future STT trends is key for interpreting the future evolution in tropospheric ozone, and the present study demonstrates the usefulness of idealized tracers to better constrain transport changes. At the same time, we emphasize that tropospheric chemistry, and in particular of the evolution of precursor emissions in each scenario, remains a major source of uncertainty.

*Author contributions.* MA wrote the draft and carried out the analyses. CO and DK contributed with additional simulations. All coauthors
contributed with discussions, expertise on the models and comments on the draft.

*Competing interests.* The authors declare no competing interests.



*Acknowledgements.* MA acknowledges funding from the Program Atracción de Talento de la Comunidad de Madrid (2016-T2/AMB-1405) and the Spanish National Project STEADY (CGL2017-83198-R). This study has been partly carried out using the high performance computing and storage facilities provided by CISL/NCAR. The EMAC simulations have been performed at the German Climate Computing Centre (DKRZ) through support from the Bundesministerium für Bildung und Forschung (BMBF). DKRZ and its scientific steering committee are

5   gratefully acknowledged for providing the HPC and data archiving resources for the consortial project ESCiMo (Earth System Chemistry integrated Modelling). We acknowledge the UK Met Office for use of the MetUM. This research was supported by the NZ Government's Strategic Science Investment Fund (SSIF) through the NIWA programme CACV. Olaf Morgenstern acknowledges funding by the New Zealand Royal Society Marsden Fund (grant 12-NIW-006). The authors wish to acknowledge the contribution of NeSI high-performance computing facilities to the results of this research. New Zealand's national facilities are provided by the New Zealand eScience Infrastructure

10  (NeSI) and funded jointly by NeSI's collaborator institutions and through the Ministry of Business, Innovation and Employment's Research Infrastructure programme (https://www.nesi.org.nz). The GEOSCCM is supported by the NASA MAP program and the high-performance computing resources were provided by the NASA Center for Climate Simulation (NCCS).



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



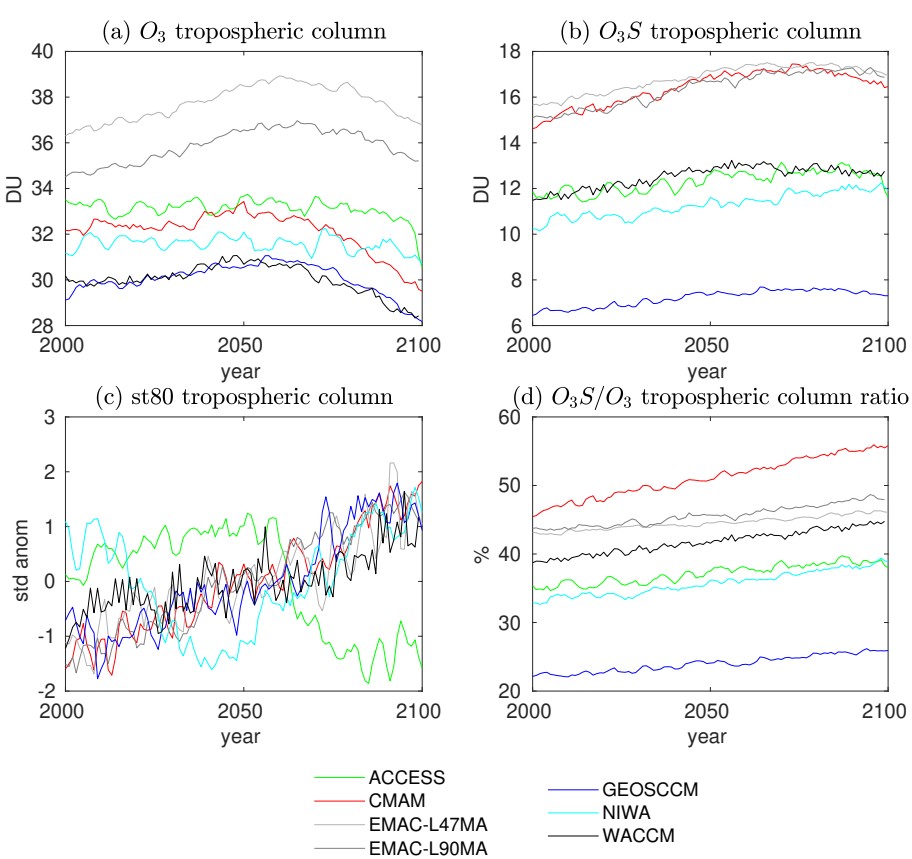

**Figure 1.** Time series of tropospheric columns of ozone (a), stratospheric ozone (b), st80 (c) and the ratio between $O_3$ and $O_3S$ (d). Ozone and stratospheric ozone are plotted in Dobson units, while st80 is shown as standardized anomalies (removing the mean and dividing by the standard deviation).



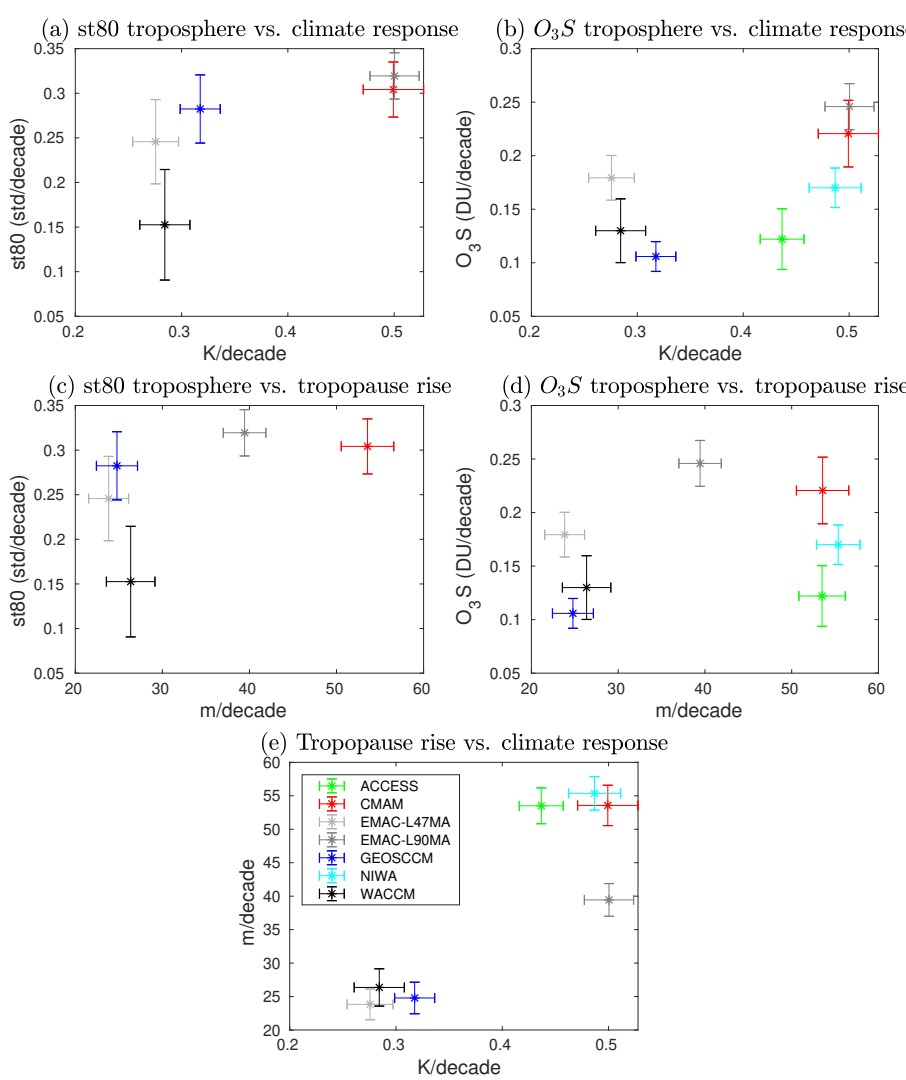

**Figure 2.** Trends over the 21st century in idealized tracers versus model climate response (a, b), evaluated as the tropical upper tropospheric temperature trends (30S-30N and 400-150 hPa), and versus globally integrated tropopause rise (c, d). Panel (e) shows tropopause rise as a function of climate response. The error bars represent the trend uncertainty calculated with a Student t test with a 95% confidence level.





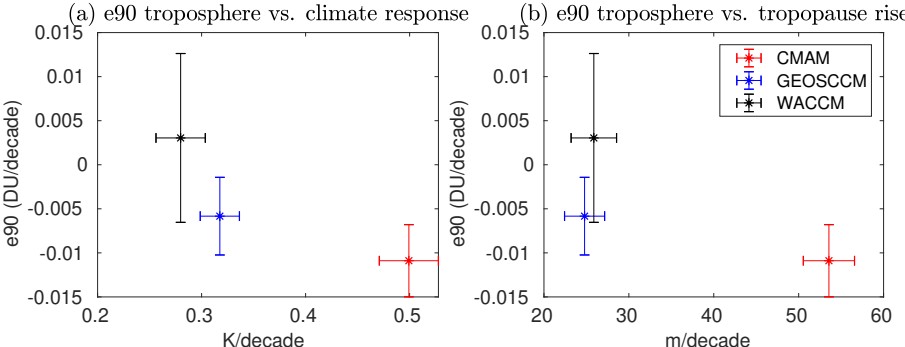

**Figure 3.** Trends over the 21st century in tropospheric e90 column versus climate response (a) and tropopause rise (b). The error bars represent the trend uncertainty with a 95% confidence level.

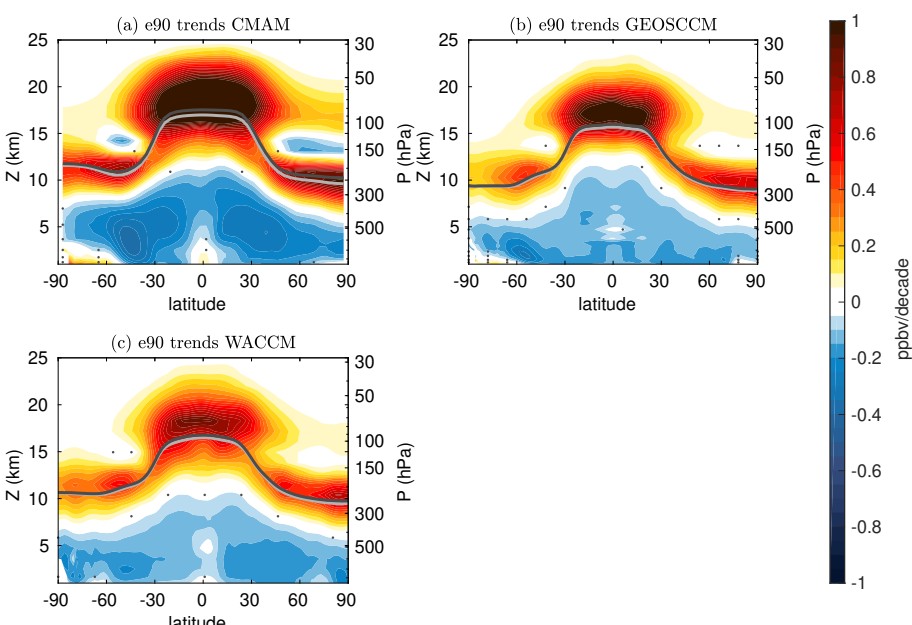

**Figure 4.** Trends over the 21st century in e90 (ppbv/decade). The dotted regions indicate where the trends are not statistically significant for a 95% confidence level. The light and dark gray lines indicate the location of the tropopause in the first and last 10 years of the 21st century, respectively. Note that Z is the log-pressure altitude $Z = Hln(p_0/p)$, with H=7 km and $p_0$=1000 hPa in this figure and in all other cross-sections.

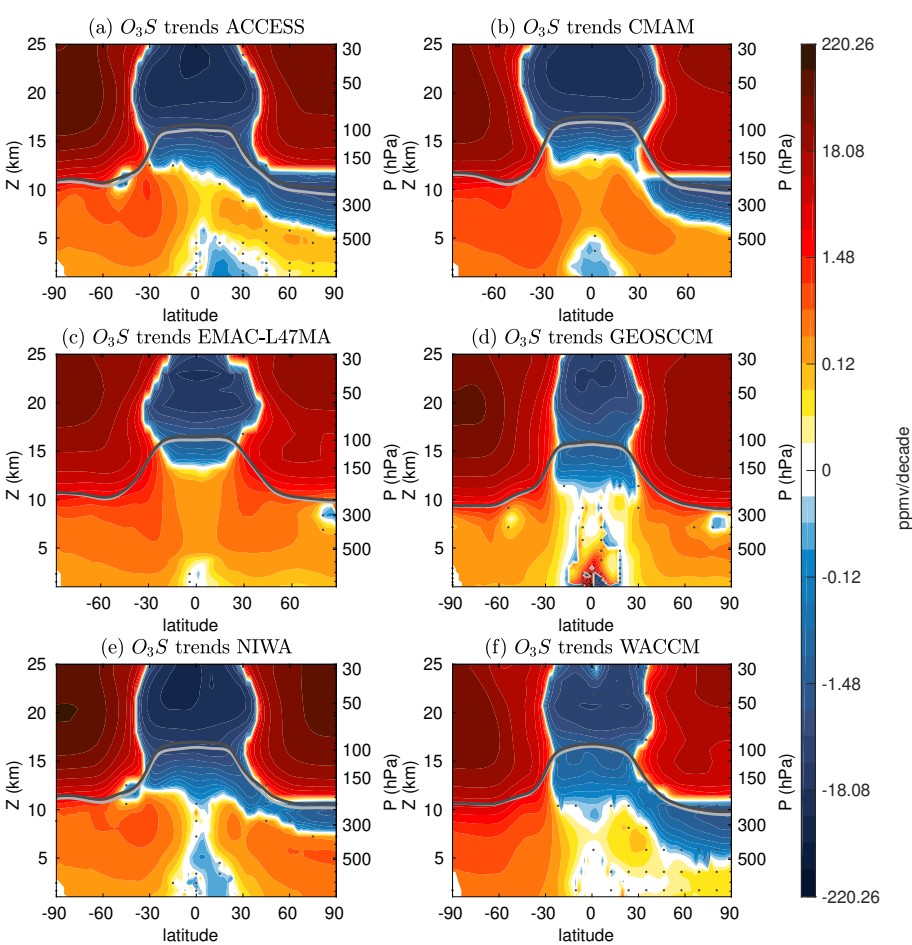

**Figure 5.** Trends over the 21st century in stratospheric ozone (ppmv/decade, in logarithmic scale). The dotted regions indicate where the trends are not statistically significant for a 95% confidence level. The light and dark gray lines indicate the location of the tropopause in the first and last 10 years of the 21st century, respectively.



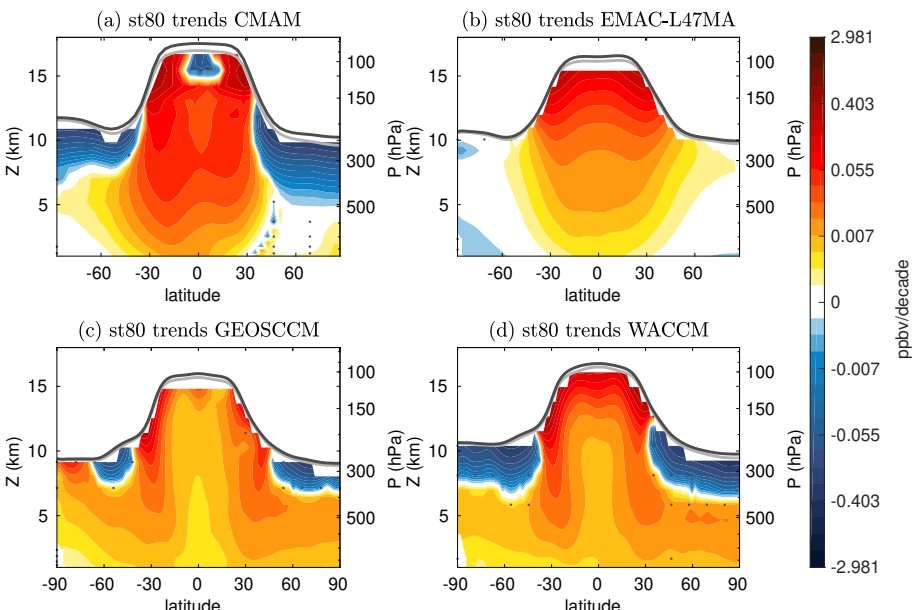

**Figure 6.** Trends over the 21st century in stratospheric tracer st80 (ppbv/day, in logarithmic scale). The dotted regions indicate where the trends are not statistically significant for a 95% confidence level. The light and dark gray lines indicate the location of the tropopause in the first and last 10 years of the 21st century, respectively.



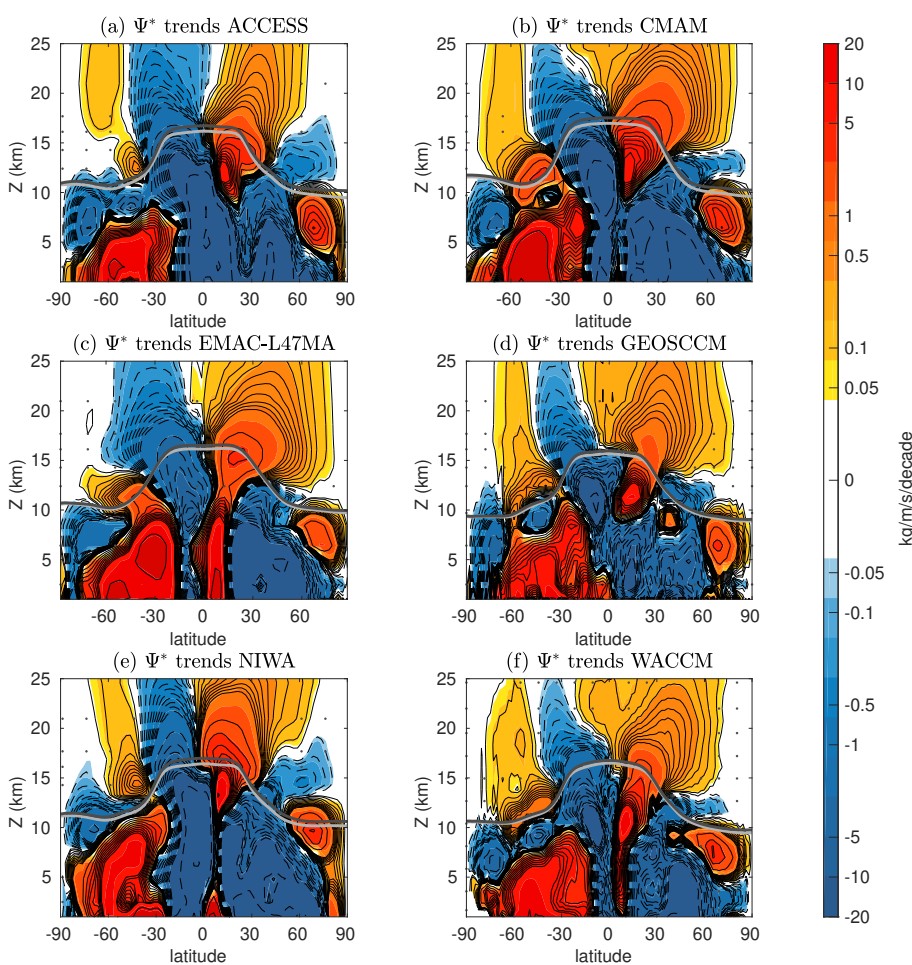

**Figure 7.** Trends over the 21st century in residual circulation streamfunction. The dotted regions indicate where the trends are not statistically significant for a 95% confidence level. The light and dark gray lines indicate the location of the tropopause in the first and last 10 years of the 21st century, respectively.

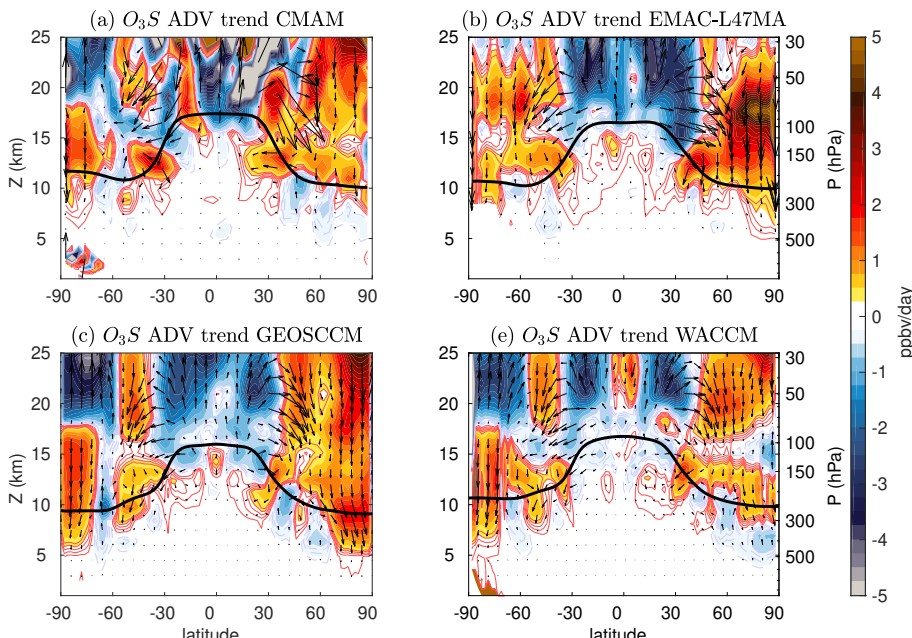

**Figure 8.** Change over the 21st century in the $O_3S$ TEM advective transport term (shading, ppbv/day) in CMAM (a), EMAC-L47MA (b), GEOSCCM (c) and WACCM (d). The arrows show the residual circulation components multiplied by the absolute value of the corresponding tracer gradient.

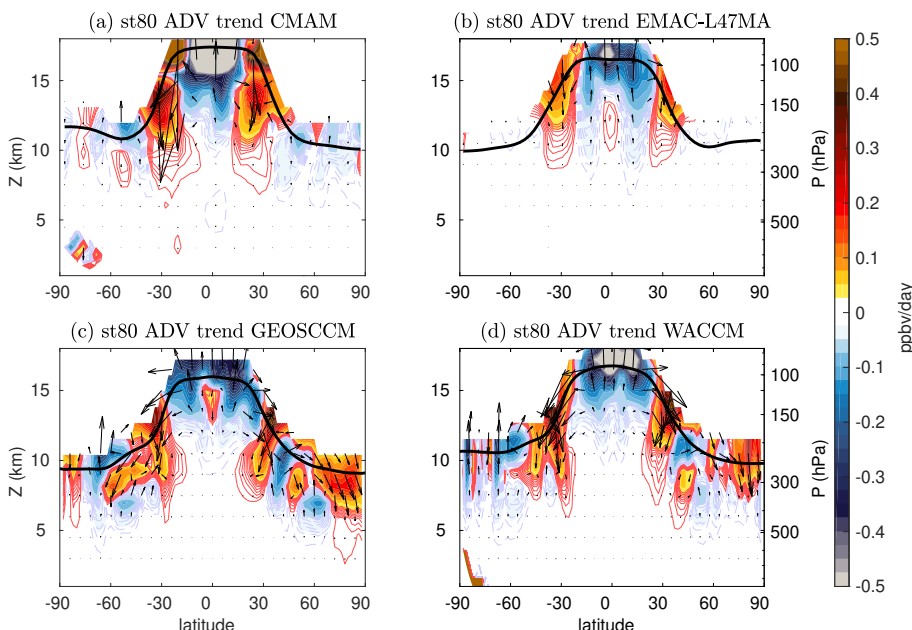

**Figure 9.** Change over the 21st century in the st80 TEM advective transport term (shading, ppbv/day) in CMAM (a), EMAC-L47MA (b), GEOSCCM (c) and WACCM (d). The arrows show the residual circulation components multiplied by the absolute value of the corresponding tracer gradient.

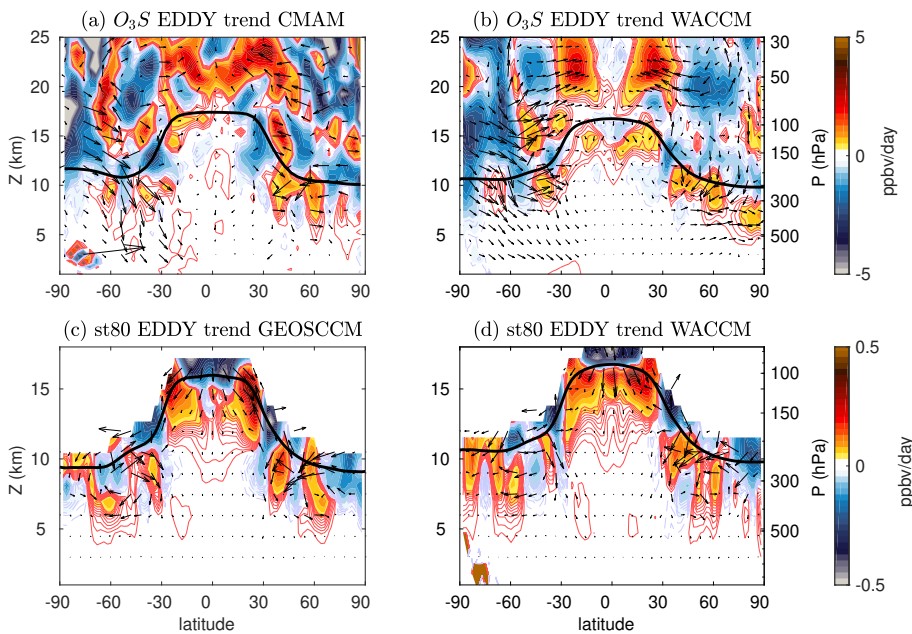

**Figure 10.** Change over the 21st century in the st80 TEM eddy transport term (shading, ppbv/day) in CMAM (a), EMAC-L47MA (b), GEOSCCM (c) and WACCM (d). Arrows denote the direction of the eddy tracer flux.

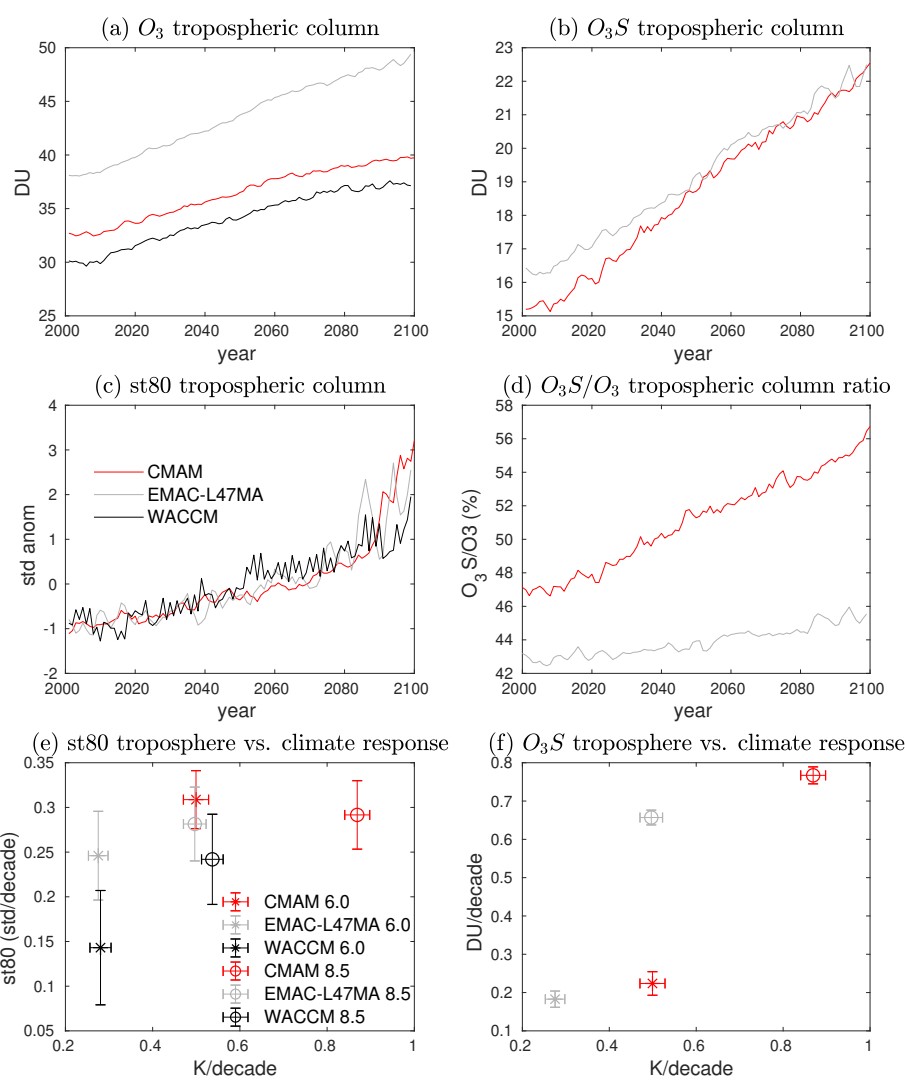

**Figure 11.** Panels a-d: Timeseries in tropospheric column tracers for the RCP8.5 scenario. Panels e and f: Trends in tropospheric column tracers versus model climate response for the RCP8.5 (circles) and the RCP6.0 (stars) emission scenarios. The error bars represent the trend uncertainty with a 95% confidence level. Note that the $O_3S$ tracer is not available in the RCP8.5 WACCM simulation.

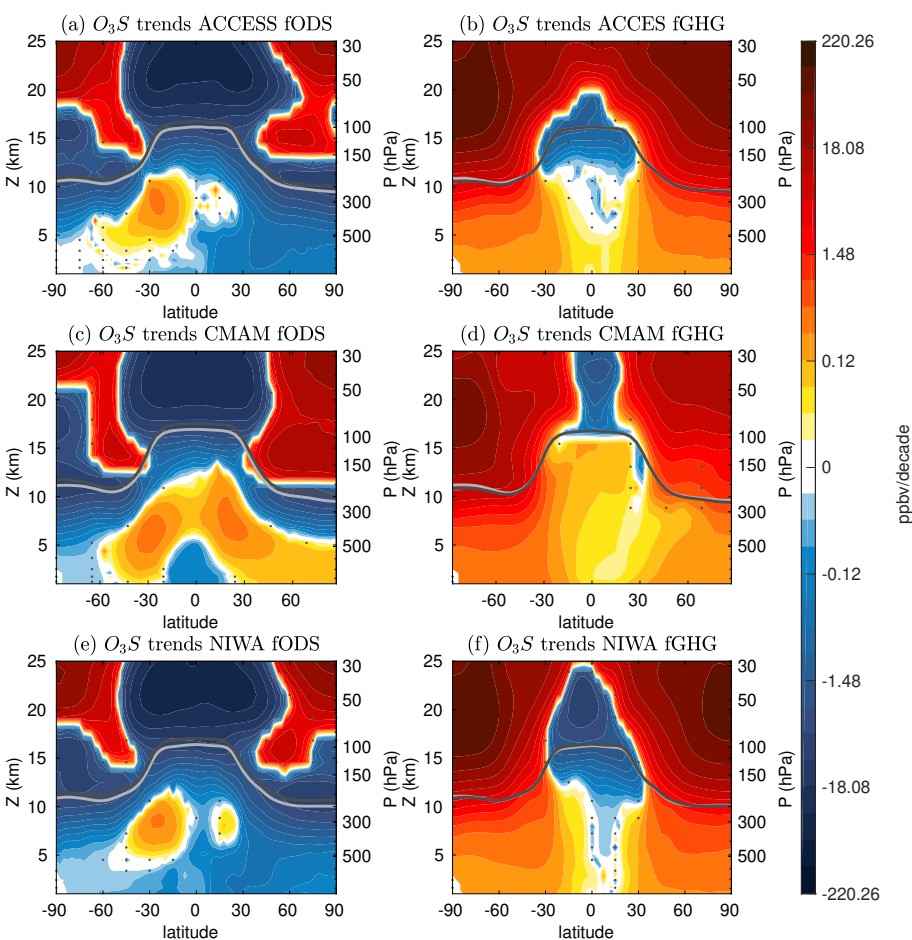

**Figure 12.** Trends over the 21st century in $O_3S$ (ppbv/decade) for the simulations with ODS (a, b) or GHG (c, d) concentrations fixed to 1960 levels in ACCESS (a, c) and CMAM (b, d). The dotted regions indicate where the trends are not statistically significant for a 95% confidence level. The light and dark gray lines indicate the location of the tropopause in the first and last 10 years of the 21st century, respectively.

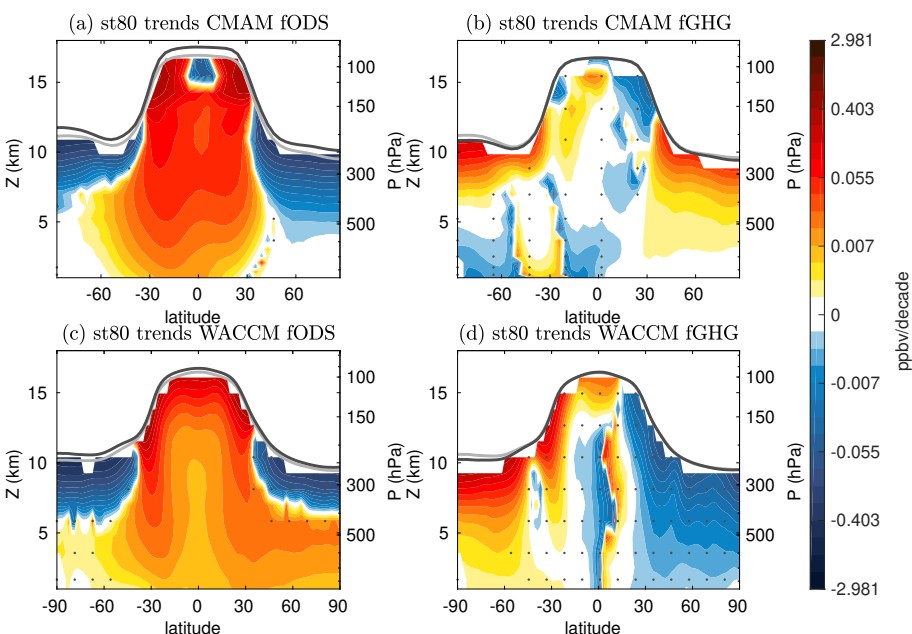

**Figure 13.** Trends over the 21st century in st80 for the simulations with ODS (a, b) or GHG (c, d) concentrations fixed to 1960 levels in CMAM (a, c) and WACCM (b, d). The dotted regions indicate where the trends are not statistically significant for a 95% confidence level. The light and dark gray lines indicate the location of the tropopause in the first and last 10 years of the 21st century, respectively.