# Peer review of "Future trends in stratosphere-to-troposphere transport in CCMI models"

_Atmospheric Chemistry and Physics, 2019_

## Referee Comment (RC1) · Anonymous Referee #1 · 7 Aug 2019

***Atmospheric Chemistry and Physics manuscript review of***:
*"Future trends in stratosphere-to-troposphere transport in CCMI models"*
*By: Abalos et al. 2019*

Overall this is nice paper that adds new insights to the literature regarding future changes in atmospheric transport. I have some questions regarding interpretation of figures, and I hope at the same time the authors can expand/clarify a bit regarding the physical mechanisms at play (mostly in the NH extratropics between 500 hPa and the lower stratosphere). I think that sorting out and discussing the physical mechanisms in this region is important because as it stands, the authors have provided convincing explanations for the stratospheric transport mechanisms, but less so for the STT that is the main topic of this paper. Nevertheless, if those issues can be addressed, then I would be happy to recommend this manuscript for publication.

**Major comments:**

Comment #1 – page 2 lines 26-30: I would be a little more careful with your language here regarding the physical mechanisms underlying STT. I say this because purely quasi-isentropic mixing is only one mechanism whereby STT occurs. For example, *cross isentropic* mixing related to transverse circulations cannot be ignored even when you are talking about mixing events that begin as PV disturbances on isentropic surfaces (e.g., Langford JGR 1999). Another way to say this is that one needs to be cognizant of the fact that the actual exchange mechanism associated with wave breaking can take on a very different flavor depending on where you are on the globe. For example, quasi-isentropic mixing (at least in my mind), generally refers to the downgradient (scale-wise) mechanical mixing of filaments that occurs *along isentropes*. That is, there is not a whole lot of *cross-isentropic* mixing taking place (i.e., largely horizontal, but not so much in the vertical). This is typically the dominant mechanism in places like in the deep subtropics where STT is occurring on or near to the 350 K surface (e.g., Waugh and Polvani 2000 or Albers et al. 2016) or say in the interior of the stratosphere in the form of the south-north "eddy mixing" portions of the BDC.

In the extratropics on the other hand, and in particular in association with the polar front, wave breaking begins with three dimensional folding/corkscrewing of the tropopause along isentropes, but in this case, a great deal of the mass exchange is associated with *vertical* turbulent mixing/erosion and diabatically induced cross-isentropic mixing of the fold itself. These are the ideas discussed in Shapiro (1980), Langford and Reid (1998), Wernli and Sprenger (2007), and Sprenger et al. (2003) (as well as the Stohl STACCATO paper you reference). Now, I'm not suggesting that you go into as much detail as the Wernli/Sprenger papers, i.e., it's probably not relevant for your paper to spend time discussing the intricacies of how wave breaking STT is manifest as folds vs. streamers vs. cutoff lows; however, I do think that you should be a little more precise with your discussion of the physical mechanisms responsible for mass exchange and the fact that in different regions of the globe, there are different processes at play.

Comment #2 – page 7 lines 15-25: I must be missing something here, because when I look at Figure 5, I do not see common behavior across all models, rather EMAC and GEOS have one behavior in the NH UTLS (positive trends), while all the other models (WACCM, CMAM, etc.) have a negative trend. This seems like a pretty notable difference that needs to be addressed.

Comment #3 – page 7 lines 15-25: This question is related to my comment #2 immediately above. In Fig. 5, the negative trends in the tropical stratosphere are easily explained via the enhancement of the BDC, so that portion of your physical explanation seems fine. However, when you state that "…*as stated above, these trends around the extratropical tropopause*….", it is unclear which explanation above you are referring to because it would seem that the rise in the tropopause (which is not particularly large) cannot alone explain the bulk of the negative trends in O3S that extend all the way down to 500 hPa in the extratropics. This would leave isentropic mixing and the residual circulation to explain the trends. I can think of a couple options here, but some of them don't seem to be consistent with your streamfunction plots in Figs. 7 and 10.

For example, your streamfunction plots show that the residual circulation accelerates coming up and out of the tropics (as expected if the deep branch of the BDC accelerates), but then there is a notable region of deceleration between 30º-80º N between 10-15 km (depending on the model). Now, I'm not sure how to label this region of negative streamfunction trend though it would seem that it could qualify as being part shallow branch and part of the lowermost portion of extratropical deep branch (perhaps this distinction is a bit ill-posed), but regardless of the what aspect of the BDC it is, the fact that it weakens could plausibly mean that less ozone is being transported downwards into the UTLS, hence helping to explain the decreasing ozone trend right at or above the tropopause. And if there is less ozone around the tropopause, then there is less ozone to be mixed via tropopause folds etc into the extratropical mid-to-upper troposphere, which would in total help explain the overall negative extratropical UTLS trend. However, all of the models have the same qualitative streamfunction trend, yet as I stated above, EMAC and GEOS do NOT show the negative O3S trend. Thus, it would seem difficult to explain the extratropical UTLS trend via the residual circulation (your Fig. 8 seems to confirm this conclusion because again, all of the models have the same qualitative advective changes, yet not all models get the same extra. UTLS O3S trend).

That leaves mixing to explain the trends. However again, the eddy transports don't (at least to my eye) seem to help explain why there is a negative trend in some models but a positive trend in others.

Please help me and other readers to understand what is physically going on here.

Comment #4 – page 9 line 11: I'm not sure I agree with your statement that Fig. 9 shows that ADV is the same in all models. I would agree that it is qualitatively the same in the UTLS between -30º-30º in all models. However, the extratropical UTLS shows two different behaviors (CMAM and EMAC vs. GEOS and WACCM). And again, this seems to point to the fact that you cannot explain the different O3S trends in the extratropical UTLS shown in Fig. 5 via advection, because while in Fig. 5 EMAC and GEOS show

similar behavior (positive trend), WACCM and CMAM show a negative trend, yet in Fig. 9, WACCM and GEOS have similar extratropical patterns (somewhat complicated, but consistent), while CMAM and EMAC show similar behavior (essentially no trend in the extratropics). Now I realize that s80 and O3S are different tracers, but given that they are both being advected via the same dynamics, it would seem that there should be some underlying commonality that can help rectify the different extratropical UTLS O3S patterns between the different models. Thus, it would help if the differences were at a minimum mentioned and hopefully the implications of these differences explained.

**Minor comments:**

Comment #1 – page 7 line 17: Where you state "*…are attributed to changes…*", instead of 'changes' can you be more precise and state what the change is? A 'decrease'?

Comment #2 – Figures: A bunch of your figures are missing pressure labels on their axis. Some figures have the labels, while other don't. Personally I find the pressure labels helpful, so perhaps you can add them for all figures?

**References:**

Langford, A. (1999). Stratosphere-troposphere exchange at the subtropical jet: Contribution to the tropospheric ozone budget at midlatitudes. *Geophysical Research Letters*, *26*(16), 2449–2452.

Shapiro, M. (1980). Turbulent mixing within tropopause folds as a mechanism for the exchange of chemical constituents between the stratosphere and troposphere. *Journal of the Atmospheric Sciences*, *37*(5), 994–1004.

Langford, A., & Reid, S. (1998). Dissipation and mixing of a small-scale stratospheric intrusion in the upper troposphere. *Journal of Geophysical Research*, *103*(D23), 31,265–31,276

Waugh, D. W., & Polvani, L. M. (2000). Climatology of intrusions into the tropical upper troposphere. *Geophysical Research Letters*, *27*(23), 3857 – 3860.

Albers, J. R., Kiladis, G. N., Birner, T., & Dias, J. (2016). Tropical upper-tropospheric potential vorticity intrusions during sudden stratospheric warmings. *Journal of the Atmospheric Sciences*, *73*(6), 2361–2384.

Wernli, H. and M. Sprenger (2007): Identification and ERA-15 Climatology of Potential Vorticity Streamers and Cutoffs near the Extratropical Tropopause, *Journal of the Atmospheric Sciences*.

Sprenger et al. (2003): Tropopause folds and cross-tropopause exchange: A global investigation based upon ECMWF analyses for the time period March 2000 to February 2001. *J. Geophys. Res*.

---

## Referee Comment (RC2) · Anonymous Referee #2 · 6 Sep 2019

Review of "Future trends in stratosphere-to-troposphere transport in CCMI models"

Recommendation: minor revision

This paper by Abalos et al. brings insights on STT as well as the overall atmospheric tracer transport in the UTLS region, and is well written. It is important to confirm the role of tropopause rise on transport in the extratropical UTLS region from multiple models with multiple idealized tracers. Also, the results highlighting the role of advective transport for subtropical STT are informative.

However, the paper does have a few places that are somewhat confusing. Below I'll list my major comments followed by other minor and technic comments. I'm happy to see the revision after the authors appropriately respond or address these concerns.

Major comments:
   (a) **The use of TEM budget analysis.**
       Besides that the advective transport shows significant contribution to the subtropical tongue of st80, I feel like the TEM budget analysis does not help that much on interpreting the spatial distribution of STT trend inferred by two stratospheric tracers (O3S and st80). Firstly, I am wondering how consistent, in terms of the spatial distribution, between the trend of tracers (unit: ppbv/decade) and the combined tracer tendency from advective-diffusive processes estimated in the TEM framework (unit: ppbv/day)? My first impression is not so much. For example, none of the advective transport or eddy mixing or combined can explain the negative trend of O3S in the NH extratropics especially the part below the tropopause. Moreover, the prominent positive trend of O3S in the SH extratropics in contrasting to the negative trend in the NH extratropics is suggested by neither advective transport nor eddy mixing of O3S. Therefore, multiple things need to be checked, which include (i) how much the trend is captured by tracer concentration differences between the present and the future, (ii) how much the difference is captured by the resolved transport approximated by the TEM framework. These checks have mostly been done in Abalos et al. (2017) so should not be problems to additionally apply to the stratospheric tracers. Also, Abalos et al. (2017) used tracer concentration difference (see their equation 2) to interpret the future trend of e90, but I am not so sure how valid it is for O3S if there is also a change in O3S lifetime $\tau$ (lifetime is fixed for e90 and st80). In sum, I think interpretation of TEM budget analysis should use more cautions if the leading spatial features of tracer trend (especially those near the tropopause) cannot be captured by this framework.

   (b) **Interpretation of stratospheric tracer trends at the extratropical tropopause.**
       The paper has shown positive trend of e90 and negative trends of stratospheric tracers over the extratropical tropopause region, except the O3S in the SH extratropics where the authors argued recovery of ozone hole matters. I highly agree with the authors this feature is associated with the upward shift of tropopause. However, I am not so sure for their additional claim on enhanced isentropic mixing on the tropopause. In my opinion, without any change in the strength of mixing at the tropopause, an upward shift of tropopause alone can already cause the increase (decrease) of tropospheric (stratospheric)

tracers in the tropopause region. Specifically, as the tropopause shifting upward, tropospheric tracers (e.g., e90) can move further upward before encountering the transport barrier by tropopause and thus more tropospheric tracers near the tropopause region which the positive trend tends to maximize in between the old and new tropopauses (see Fig. 4). By contrast, as the tropopause shifting upward, downward transport of stratospheric tracers encounters earlier with the tropopause barrier, and thus less stratospheric tracers near the tropopause region with the negative trend also maximizing in between the old and new tropopauses, as shown in Figs. 5 and 6. The enhancement of isentropic mixing on the tropopause could indeed amplify this effect, but given the fact that models are not showing consistent results about the eddy mixing component (briefly noted in the manuscript) plus the results of eddy mixing component are much more noisy, I don't think a strong conclusion on enhanced isentropic mixing on the tropopause can be made. Finally, as noted earlier in (a), neither advective transport nor eddy mixing seem to reflect the prominent negative trend of stratospheric tracers in the NH extratropics, particularly the part below the tropopause. Therefore, I suspect that the TEM budget analysis may not show up the effect of tropopause rise on extratratropic tracer transport.

(c) **Lack of mechanism interpretation on inter-model differences of STT.**
The authors give some good examples of comparing trends of tropospheric-column averaged tracer concentration for O3S and st80 in Section 3.1 and 5 to highlight the inter-model differences, which is "one great merit" of looking at inter-model comparison project. However, when coming across the discussion of mechanism in Section 4, none of these inter-model differences are noted again, so is the spatial distribution in Section 3.2. It is good to focus on common features that are supported by most of models, but the inter-model differences could also bring some interesting insights. For example, models like CMAM show larger tropospheric appearance of st80 than models like WACCM and GEOSCCM, which are likely due to stronger subtropical tongue in CMAM than those in WACCM/GEOSCCM and therefore link to stronger lower BDC and upper HC overturning in CMAM than those in WACCM/GEOSCCM (see Fig. 7). This again highlights the importance of advective transport for STT of st80. The inter-model differences in O3S are more complicated but a brief discussion may be helpful.

Minor (and technic) comments:
P1: Institutions 4 and 5 should switch place.

P6L15-P6L19: From later results, it seems that the tropopause rise is more related to variations in STT over the extratropics instead of the global STT shown by tropospheric burden of these stratospheric tracers. The global STT is likely controlled by other processes (e.g., subtropical tongue due to overturning circulation in the UTLS for st80). In short, I am not surprised that the correlation is weaker for tropopause rise than climate response and I doubt how confident the authors can argue tropopause rise act as an important mediator for the global budget.

P7L34-P8L1: The authors noted some differences about spatial distribution between O3S and st80. I think this should be highlighted more often in the manuscript to warn readers that interpretation of O3S should use more cautions as both variations in stratospheric chemistry and

source distributions could yield different behaviors from st80. Also, I suggest the authors to insert cross-section maps for climatological O3S and st80 distribution (either a new figure or superimposed in existing figures as contours) so that readers can have a better idea on how the future changes in tracers compare to the climatological distribution.

P8L2-P8L5: I think st80 in upper-troposphere deep tropics can also be interpreted by later results of advective transport and eddy mixing. From Figs. 9 and 10, the advective transport of st80 in deep tropics generally shows negative trend while the eddy mixing shows positive trend. The eddy mixing component seems to have a larger trend than the advective transport so that the net compensation outcome shows the positive trend. For CMAM model in which the negative trend by advective transport is so strong in deep tropics that cannot be fully compensated by eddy mixing shows a net outcome of local negative trend. In sum, I agree with the authors that variations of st80 in deep tropics are related to enhanced diffusion on the tropopause but not quite sure whether the tropopause altitude playing a role here. As mentioned in the major comments, in my opinion, tropopause rise works better for extratropical STT variations which its influence seems not to be captured by the TEM diagnostics.

P8L14: las -> last

P8L15: Polvani et al. 2019 -> Polvani et al. (2019)

P9L2-P9L3: This part reads so similar to earlier part of ozone recovery-related variations in residual circulation, so it confuses initially. I suggest to diffrentiate at the beginning about the double effects of ozone recovery on STT of O3S: (i) weaken the downwelling of residual circulation leading to less polar O3S accumulation by transport, and (ii) increase polar ozone concentrations. Effects of (i) dominates above 20 km so O3S shows negative trend while effects of (ii) surpass below 20 km so that O3S shows positive trend suggesting stronger STT of O3S at polar regions.

P9L12-P9L13: Although both O3S and st80 highlighting the subtropical tongue for transport in the UTLS region, there are some differences about their advective transport: (i) transport in the deep tropics which is likely due to differences in source, and (ii) subtropical tongue seems to intrude more vertically as for st80 than O3S. Do you have an idea on why is so?

P9L32-P10L1: As noted in the major comments, I suspect how strong this conclusion can be given that the TEM diagnostics seems to fail to capture the effects of tropopause rise on STT.

P10L26-P10L29: Would this be clearer if additional lines for the corresponding RCP6.0 cases are added in Fig.11(a-d)?

fGHG vs fODS: I am interested in seeing how much the addition of fGHG+fODS can explain the full trend seen in Figs. 5 and 6. Also, I think for both O3S and st80, the fODS explains more on the full trend response of STT than fGHG, which could be pointed out at the beginning of P11L14.

P11L23: is "the" strongest

P11L24-P11L25: Should these cross-references be Fig. 12?

P12L5: Polvani et al. 2019 -> Polvani et al. (2019)

P13L10: this region -> the extratropical lower stratosphere

---

## Author Comment (AC1) · 10 Feb 2020

Response to the reviewers

We thank reviewer 1 for their very insightful and constructive comments. We are certain that we substantially improved the quality of the paper thanks to this revision. We provide below a point-by-point response. The page and lines mentioned in the responses refer to the tracked-changes version of the revised manuscript.

Major comments

1- Comment #1 – page 2 lines 26-30: I would be a little more careful with your language here regarding the physical mechanisms underlying STT. I say this because purely quasi-isentropic mixing is only one mechanism whereby STT occurs. For example, cross isentropic mixing related to transverse circulations cannot be ignored even when you are talking about mixing events that begin as PV disturbances on isentropic surfaces (e.g., Langford JGR 1999). Another way to say this is that one needs to be cognizant of the fact that the actual exchange mechanism associated with wave breaking can take on a very different flavor depending on where you are on the globe. For example, quasi-isentropic mixing (at least in my mind), generally refers to the downgradient (scale-wise) mechanical mixing of filaments that occurs along isentropes. That is, there is not a whole lot of cross-isentropic mixing taking place (i.e., largely horizontal, but not so much in the vertical). This is typically the dominant mechanism in places like in the deep subtropics where STT is occurring on or near to the 350 K surface (e.g., Waugh and Polvani 2000 or Albers et al. 2016) or say in the interior of the stratosphere in the form of the south-north "eddy mixing" portions of the BDC.
In the extratropics on the other hand, and in particular in association with the polar front, wave breaking begins with three dimensional folding/corkscrewing of the tropopause along isentropes, but in this case, a great deal of the mass exchange is associated with vertical turbulent mixing/erosion and diabatically induced cross-isentropic mixing of the fold itself. These are the ideas discussed in Shapiro (1980), Langford and Reid (1998), Wernli and Sprenger (2007), and Sprenger et al. (2003) (as well as the Stohl STACCATO paper you reference). Now, I'm not suggesting that you go into as much detail as the Wernli/Sprenger papers, i.e., it's probably not relevant for your paper to spend time discussing the intricacies of how wave breaking STT is manifest as folds vs. streamers vs. cutoff lows; however, I do think that you should be a little more precise with your discussion of the physical mechanisms responsible for mass exchange and the fact that in different regions of the globe, there are different processes at play.

Thank you for this nice overview summary of STT processes. We rewrote this Introduction paragraph, clarifying the different processes that take place in subtropics and extratropics (P2 L26-34).

2- Comment #2 – page 7 lines 15-25: I must be missing something here, because when I look at Figure 5, I do not see common behavior across all models, rather EMAC and GEOS have one behavior in the NH UTLS (positive trends), while all the other models (WACCM, CMAM, etc.) have a negative trend. This seems like a pretty notable difference that needs to be addressed.

This is true, this difference should be highlighted. We included this (P7L29-31), and we also mentioned that this is probably due to a larger contribution from ozone recovery (ODS) that cancels

the negative trends expected from tropopause rise (GHG) in these two models, as can be concluded from Fig. 11, even though the output for these two models in particular is not available.

Comment #3 – page 7 lines 15-25: This question is related to my comment #2 immediately above. In Fig. 5, the negative trends in the tropical stratosphere are easily explained via the enhancement of the BDC, so that portion of your physical explanation seems fine. However, when you state that "...as stated above, these trends around the extratropical tropopause....", it is unclear which explanation above you are referring to because it would seem that the rise in the tropopause (which is not particularly large) cannot alone explain the bulk of the negative trends in O3S that extend all the way down to 500 hPa in the extratropics. This would leave isentropic mixing and the residual circulation to explain the trends. I can think of a couple options here, but some of them don't seem to be consistent with your streamfunction plots in Figs. 7 and 10.

For example, your streamfunction plots show that the residual circulation accelerates coming up and out of the tropics (as expected if the deep branch of the BDC accelerates), but then there is a notable region of deceleration between 30o-80o N between 10-15 km (depending on the model). Now, I'm not sure how to label this region of negative streamfunction trend though it would seem that it could qualify as being part shallow branch and part of the lowermost portion of extratropical deep branch (perhaps this distinction is a bit ill-posed), but regardless of the what aspect of the BDC it is, the fact that it weakens could plausibly mean that less ozone is being transported downwards into the UTLS, hence helping to explain the decreasing ozone trend right at or above the tropopause. And if there is less ozone around the tropopause, then there is less ozone to be mixed via tropopause folds etc into the extratropical mid-to-upper troposphere, which would in total help explain the overall negative extratropical UTLS trend. However, all of the models have the same qualitative streamfunction trend, yet as I stated above, EMAC and GEOS do NOT show the negative O3S trend. Thus, it would seem difficult to explain the extratropical UTLS trend via the residual circulation (your Fig. 8 seems to confirm this conclusion because again, all of the models have the same qualitative advective changes, yet not all models get the same extra. UTLS O3S trend).

That leaves mixing to explain the trends. However again, the eddy transports don't (at least to my eye) seem to help explain why there is a negative trend in some models but a positive trend in others.

Please help me and other readers to understand what is physically going on here.

Our argument is that changes in tropopause height, small as they are on average, lead to substantial changes in the tracer concentrations. In Abalos et al. 2017 JAS we showed this to be the case for the tropospheric tracer e90, and here we present the same argument for the stratospheric tracers o3S and st80. To highlight this point, we now mention that the band of negative trends in the extratropical UTLS disappears when tropopause-relative altitude coordinate is used. This strong influence of tropopause rise can be explained in two ways (i.e., through two mechanisms which we think are acting together). First, concomitant with the tropopause rise there is an (upward) expansion of the troposphere, which implies that the air around the tropopause becomes more tropospheric-like, and consistently the concentrations of stratospheric tracer decrease. In addition, we argue, following Abalos et al. 2017 JAS, that the upward shift of the tropopause is linked to changes in static stability which in turn modify the wave propagation and dissipation conditions, resulting in modified mixing strength. We do observe some signal of this enhanced mixing in the TEM eddy transport term, but the lack of consistency among models, and the fact that negative trends extend to lower levels than the mixing trends, prevents us to be conclusive about this last mechanism. Thus, we have lowered

the tone attributing these trends to changes in mixing and mentioned the first mechanism, also following suggestions of the other reviewer. These changes can be found in several places in the paper: P7L14-21, P10L34-35, P15L1-9. Regarding the difference between models, it is likely due to different degrees of cancellation between the opposite effects of ODS and GHG in different models, as mentioned in the response to your comment #2.

Comment #4 – page 9 line 11: I'm not sure I agree with your statement that Fig. 9 shows that ADV is the same in all models. I would agree that it is qualitatively the same in the UTLS between -30o-30o in all models. However, the extratropical UTLS shows two different behaviors (CMAM and EMAC vs. GEOS and WACCM). And again, this seems to point to the fact that you cannot explain the different O3S trends in the extratropical UTLS shown in Fig. 5 via advection, because while in Fig. 5 EMAC and GEOS showsimilar behavior (positive trend), WACCM and CMAM show a negative trend, yet in Fig. 9, WACCM and GEOS have similar extratropical patterns (somewhat complicated, but consistent), while CMAM and EMAC show similar behavior (essentially no trend in the extratropics). Now I realize that s80 and O3S are different tracers, but given that they are both being advected via the same dynamics, it would seem that there should be some underlying commonality that can help rectify the different extratropical UTLS O3S patterns between the different models. Thus, it would help if the differences were at a minimum mentioned and hopefully the implications of these differences explained.

We agree with the reviewer that this should be explained. The issue here is that the st80 climatology in the lower stratosphere in CMAM and EMAC is different from the other of the models, as can be seen in Orbe et al. 2018 ACP, supplementary material Figure S2. More specifically, while GEOSCCM and WACCM present positive st80 trends in the extratropical lower stratosphere (as expected from accelerated BDC downwelling), CMAM and EMAC present negative trends in this region. The enhanced downward advective transport of st80 into the polar troposphere found in GEOSCCM and WACCM is due to the increasing tracer concentrations in the lowermost stratosphere (i.e., a larger reservoir of st80 to be transported downward). The same mechanism applies to O3S: the accumulation of O3S in the lower stratosphere leads to enhanced downward advective transport into the troposphere. All the models show this consistent behavior for O3S (Fig. 7). We have explained this issue in more detail on P10 L8-15.

Minor comments

Comment #1 – page 7 line 17: Where you state "...are attributed to changes...", instead of 'changes' can you be more precise and state what the change is? A 'decrease'?

Consistent with our response to comment #3, we have lowered the tone on the attribution to mixing and now we changed the sentence to: "As stated above, the negative trends in the extratropical UTLS are a fingerprint of the rise of the tropopause."

Comment #2 – Figures: A bunch of your figures are missing pressure labels on their axis. Some figures have the labels, while other don't. Personally I find the pressure labels helpful, so perhaps you can add them for all figures?

We added pressure levels on all panels of Figs. 7, 8, 9, 10.

References:

Langford, A. (1999). Stratosphere-troposphere exchange at the subtropical jet: Contribution to the tropospheric ozone budget at midlatitudes. Geophysical Research Letters, 26(16), 2449–2452.

Shapiro, M. (1980). Turbulent mixing within tropopause folds as a mechanism for the exchange of chemical constituents between the stratosphere and troposphere. Journal of the Atmospheric Sciences, 37(5), 994–1004.

Langford, A., & Reid, S. (1998). Dissipation and mixing of a small-scale stratospheric intrusion in the upper troposphere. Journal of Geophysical Research, 103(D23), 31,265–31,276

Waugh, D. W., & Polvani, L. M. (2000). Climatology of intrusions into the tropical upper troposphere. Geophysical Research Letters, 27(23), 3857 – 3860.

Albers, J. R., Kiladis, G. N., Birner, T., & Dias, J. (2016). Tropical upper-tropospheric potential vorticity intrusions during sudden stratospheric warmings. Journal of the Atmospheric Sciences, 73(6), 2361–2384.

Wernli, H. and M. Sprenger (2007): Identification and ERA-15 Climatology of Potential Vorticity Streamers and Cutoffs near the Extratropical Tropopause, Journal of the Atmospheric Sciences.

Sprenger et al. (2003): Tropopause folds and cross-tropopause exchange: A global investigation based upon ECMWF analyses for the time period March 2000 to February 2001. J. Geophys. Res.

We acknowledge the careful and constructive review which has contributed to notably improve the paper. We provide a point-by-point response below.

Major comments:

**(a) The use of TEM budget analysis.**
Besides that the advective transport shows significant contribution to the subtropical tongue of st80, I feel like the TEM budget analysis does not help that much on interpreting the spatial distribution of STT trend inferred by two stratospheric tracers (O3S and st80). Firstly, I am wondering how consistent, in terms of the spatial distribution, between the trend of tracers (unit: ppbv/decade) and the combined tracer tendency from advective-diffusive processes estimated in the TEM framework (unit: ppbv/day)? My first impression is not so much. For example, none of the advective transport or eddy mixing or combined can explain the negative trend of O3S in the NH extratropics especially the part below the tropopause. Moreover, the prominent positive trend of O3S in the SH extratropics in contrasting to the negative trend in the NH extratropics is suggested by neither advective transport nor eddy mixing of O3S. Therefore, multiple things need to be checked, which include (i) how much the trend is captured by tracer concentration differences between the present and the future, (ii) how much the difference is captured by the resolved transport approximated by the TEM framework. These checks have mostly been done in Abalos et al. (2017) so should not be problems to additionally apply to the stratospheric tracers. Also, Abalos et al. (2017) used tracer concentration difference (see their equation 2) to interpret the future trend of e90, but I am not so sure how valid it is for O3S if there is also a change in O3S lifetime τ (lifetime is fixed for e90 and st80). In sum, I think interpretation of TEM budget analysis should use more cautions if the leading spatial features of tracer trend (especially those near the tropopause) cannot be captured by this framework.

Thank you for this comment. Regarding (i): we have checked that the trends and the future minus past difference in the tracer concentrations are highly consistent (not shown). Regarding (ii): we do not attempt here to provide a quantitative estimate of the contribution from each term to the net tracer trends. In order to do this, we should include all the other terms in the balance in addition to the resolved transport terms (advection and eddy transport): non-resolved transport terms (convection, diffusion, transport by subgrid-scale waves) and the chemical tendency term (likely important for O3S). Still, there would be a residual that prevents closing the budget, due to the numerical issues such as those listed in Abalos et al 2017 JAS, and their equivalent for other models. Nevertheless, the two resolved transport terms give valuable information on the effects of these two types of transport mechanisms on the tracer trends. In particular, the advective transport term helps illustrate the effects of the residual circulation trends on the tracers STT, which we consider a key point of this paper. These effects would be hard to mentally picture from the trends in the circulation alone (Fig. 7). In addition to the subtropical tongues mentioned by the reviewer, the NH polar downwelling trends are evidenced, which we have now emphasized in the revised version of the paper.

Finally, as the reviewer correctly states, we cannot apply Eq. (2) from Abalos et al. 2017 to O3S, given that it has time-varying chemical tendencies. This is why we have expressed the TEM trends in ppbv/day (difference between transport terms in the future minus past), instead of converting them to ppbv, as was done in the 2017 paper for e90.

We are confident that the explanations included in the revised manuscript ensure that the results regarding the TEM terms are interpreted correctly, accounting for their limitations. See P11 L5-10.

**(b) Interpretation of stratospheric tracer trends at the extratropical tropopause.**
The paper has shown positive trend of e90 and negative trends of stratospheric tracers over the extratropical tropopause region, except the O3S in the SH extratropics where the authors argued recovery of ozone hole matters. I highly agree with the authors this feature is associated with the upward shift of tropopause. However, I am not so sure for their additional claim on enhanced isentropic mixing on the tropopause. In my opinion, without any change in the strength of mixing at the tropopause, an upward shift of tropopause alone can already cause the increase (decrease) of tropospheric (stratospheric)tracers in the tropopause region. Specifically, as the tropopause shifting upward, tropospheric tracers (e.g., e90) can move further upward before encountering the transport barrier by tropopause and thus more tropospheric tracers near the tropopause region which the positive trend tends to maximize in between the old and new tropopauses (see Fig. 4). By contrast, as the tropopause shifting upward, downward transport of stratospheric tracers encounters earlier with the tropopause barrier, and thus less stratospheric tracers near the tropopause region with the negative trend also maximizing in between the old and new tropopauses, as shown in Figs. 5 and 6. The enhancement of isentropic mixing on the tropopause could indeed amplify this effect, but given the fact that models are not showing consistent results about the eddy mixing component (briefly noted in the manuscript) plus the results of eddy mixing component are much more noisy, I don't think a strong conclusion on enhanced isentropic mixing on the tropopause can be made. Finally, as noted earlier in (a), neither advective transport nor eddy mixing seem to reflect the prominent negative trend of stratospheric tracers in the NH extratropics, particularly the part below the tropopause. Therefore, I suspect that the TEM budget analysis may not show up the effect of tropopause rise on extratratropic tracer transport.

We agree with the reviewer that we have probably overstated the role of changes in mixing on the extratropical UTLS trends, given the uncertainty in this term. We understand that there is likely a contribution directly from the fact that the troposphere expands. To address this comment, we have restated the interpretations in the paper on the extratropical UTLS trends, stressing the uncertainties in the mixing term and including this second mechanism. Moreover, we have emphasized the tropopause rise as a key factor for the extratropical UTLS trends, since this feature is the driver behind both mechanisms. These changes are found in several parts of the manuscript (P7L14-21, P10L34-35, P15L1-9).

**(c) Lack of mechanism interpretation on inter-model differences of STT.**
The authors give some good examples of comparing trends of tropospheric-column averaged tracer concentration for O3S and st80 in Section 3.1 and 5 to highlight the inter-model differences, which is "one great merit" of looking at inter-model comparison project. However, when coming across the discussion of mechanism in Section 4, none of these inter-model differences are noted again, so is the spatial distribution in Section 3.2. It is good to focus on common features that are supported by most of models, but the inter-model differences could also bring some interesting insights. For example, models like CMAM show larger tropospheric appearance of st80 than models like WACCM and GEOSCCM, which are likely due to stronger subtropical tongue in CMAM than those in WACCM/GEOSCCM and therefore link to stronger lower BDC and upper HC overturning in CMAM than those in WACCM/GEOSCCM (see Fig. 7). This again highlights the importance of advective transport for STT of st80. The inter-model differences in O3S are more complicated but a brief discussion may be helpful.

We agree with the reviewer that the inter-model spread can be used to extract information on the mechanisms, and this is one merit of multi-model studies. Accordingly, we decided to remove the scatter plots against tropopause altitude because they were not providing any information on the processes (see response to next comment). In their place, we show now plots of STT against tropical upwelling trends. In this way, we use the inter-model spread to illustrate the influence of the BDC on STT, stressing this key point of the paper. Note that we also merged the figure for the stratospheric tracers with that for e90 (old Figs. 2 and 3) and included them in the new Fig. 2, in order to reduce the number of figures from 13 to 12. See related discussion on P6 L18-36.

Minor (and technic) comments:

P1: Institutions 4 and 5 should switch place.

Done, thank you for noticing the error.

P6L15-P6L19: From later results, it seems that the tropopause rise is more related to variations in STT over the extratropics instead of the global STT shown by tropospheric burden of these stratospheric tracers. The global STT is likely controlled by other processes (e.g., subtropical tongue due to overturning circulation in the UTLS for st80). In short, I am not surprised that the correlation is weaker for tropopause rise than climate response and I doubt how confident the authors can argue tropopause rise act as an important mediator for the global budget.

We agree with the reviewer on this point. This comment, together with Major comment (c), led us to rethink the figures of the paper as described in the response to comment (c). Indeed, the old Fig. 2 showed the negative result that tropopause altitude trends are not correlated with global STT trends. The changes introduced in Fig. 2 allow to highlight two important results of the paper: 1) models with larger climate sensitivity present larger STT trends, and 2) STT trends are connected to BDC trends.

P7L34-P8L1: The authors noted some differences about spatial distribution between O3S and st80. I think this should be highlighted more often in the manuscript to warn readers that interpretation of O3S should use more cautions as both variations in stratospheric chemistry andsource distributions could yield different behaviors from st80. Also, I suggest the authors to insert cross-section maps for climatological O3S and st80 distribution (either a new figure or superimposed in existing figures as contours) so that readers can have a better idea on how the future changes in tracers compare to the climatological distribution.

Thank you for the suggestion. We have highlighted these differences more often in the manuscript and we added climatology contours in Figs. 3, 4 and 5.

P8L2-P8L5: I think st80 in upper-troposphere deep tropics can also be interpreted by later results of advective transport and eddy mixing. From Figs. 9 and 10, the advective transport of st80 in deep tropics generally shows negative trend while the eddy mixing shows positive trend. The eddy mixing component seems to have a larger trend than the advective transport so that the net compensation outcome shows the positive trend. For CMAM model in which the negative trend by advective transport is so strong in deep tropics that cannot be fully compensated by eddy mixing shows a net outcome of local negative trend. In sum, I agree with the authors that variations of st80 in deep tropics are related to enhanced diffusion on the tropopause but not quite sure whether the tropopause altitude playing a role here. As mentioned in the major comments, in my opinion, tropopause rise works better for extratropical STT variations which its influence seems not to be captured by the TEM diagnostics.

We agree with the reviewer that it is not clear that tropopause altitude is the only factor leading to differences in deep tropics st80 trends among models. However, we are not confident that the TEM analyses help us understand the model spread. In fact, as mentioned in the response to your comment #1, we prefer not making claims on relative roles of the two transport terms on the tracer trends, since we do not close the budget. On the other hand, looking at the climatological concentrations of st80 now plotted in Fig. 5, it becomes evident that the two models with stronger trends in the deep tropics (CMAM and EMAC) are also those that also have larger climatological concentrations in this region. This suggests that in these models there is more cross-tropopause diffusion of the tracer into the troposphere. The reason for this could be the different transport/diffusion properties of the models, though a higher tropical tropopause, closer to the 80 hPa source of the tracer, likely contributes to magnify this effect. We have clarified this in the manuscript (P8 L20-25).

P8L14: las -> last

Changed.

P8L15: Polvani et al. 2019 -> Polvani et al. (2019)

Changed.

P9L2-P9L3: This part reads so similar to earlier part of ozone recovery-related variations in residual circulation, so it confuses initially. I suggest to diffrentiate at the beginning about the double effects of ozone recovery on STT of O3S: (i) weaken the downwelling of residual circulation leading to less polar O3S accumulation by transport, and (ii) increase polar ozone concentrations. Effects of (i) dominates above 20 km so O3S shows negative trend while effects of (ii) surpass below 20 km so that O3S shows positive trend suggesting stronger STT of O3S at polar regions.

Changed, thank you.

P9L12-P9L13: Although both O3S and st80 highlighting the subtropical tongue for transport in the UTLS region, there are some differences about their advective transport: (i) transport in the deep tropics which is likely due to differences in source, and (ii) subtropical tongue seems to intrude more vertically as for st80 than O3S. Do you have an idea on why is so?

Regarding (i), we think it is likely due to differences in source regions, see response to your previous comment about P8L2-P8L5. Regarding (ii), considering that the advective term is -v* dX/dy – w*dX/dz, this difference has to be due to the different vertical/meridional gradients in the two tracers, since the residual circulation is the same. Specifically, for st80 the vertical term dominates over the meridional more than for O3S. However, we agree that this is not obvious from simply looking at the climatological contours in Figs. 4 and 5.

P9L32-P10L1: As noted in the major comments, I suspect how strong this conclusion can be given that the TEM diagnostics seems to fail to capture the effects of tropopause rise on STT.

We have included the direct effect of the tropospheric expansion argued by the referee as an additional driver of the extratropical UTLS trends.

P10L26-P10L29: Would this be clearer if additional lines for the corresponding RCP6.0 cases

are added in Fig.11(a-d)?

Yes, definitely, thank you for the suggestion. As we added them, we realizedthat, in order to compare runs of the same model it is best to compare the st80 without standardizing, to avoid losing information. So we changed this, and now all the models show a stronger STT trends for the more extreme scenario for both tracers, as expected. By having both curves in the same plot, we also realized that the change in stratospheric-to-total ozone ratio is not significant, and we mentioned this in the manuscript. Overall the interpretation of this figure is much clearer now.

fGHG vs fODS: I am interested in seeing how much the addition of fGHG+fODS can explain the full trend seen in Figs. 5 and 6. Also, I think for both O3S and st80, the fODS explains more on the full trend response of STT than fGHG, which could be pointed out at the beginning of P11L14.

We would like to clarify that these runs are not exactly additive, because they are not single forcing runs, but all forcing-minus-one runs. Adding them would imply adding twice some effects (such as aerosol, for instance). Nevertheless, it is insightful to examine the relative contribution of each single forcing (GHG and ODS) to the total trends, which can be done by comparing the timeseries, as shown in Fig. R1. From this figure it is clear that, in the case of st80, climate change is responsible for the trend in STT, and ODS do not play any significant role. In the case of O3S, the opposite is true for ACCESS and NIWA, i.e., ozone recovery completely dominates over the effect of GHG. In CMAM however, both effects have similar magnitude. These are indeed interesting results and we have included them in the paper thanks to the referee's suggestion. P13 L9-12.

[Figure]

Figure R1. Timeseries of st80 (a) and o3s (b) tropospheric columns for the REF-C2 runs (blue) compared to tehe SEN-C2-fODS (red) and SEN-C2-fGHG (green). The different symbols represent different models: CMAM (circles), WACCM (stars), ACCESS (crosses), NIWA (triangles). In both panels are represented anomalies with respect to the average of the first ten years.

P11L23: is "the" strongest

Changed.

P11L24-P11L25: Should these cross-references be Fig. 12?

Yes, thank you. Changed.

P12L5: Polvani et al. 2019 -> Polvani et al. (2019)

Changed.

P13L10: this region -> the extratropical lower stratosphere

This sentence is changed in the new version.